DOI: 10.1038/s41467-017-00211-5　　**OPEN**

# Alkaline earth metal vanadates as sodium-ion battery anodes

Xiaoming Xu[1], Chaojiang Niu[1], Manyi Duan[2], Xuanpeng Wang[1], Lei Huang[1], Junhui Wang[1], Liting Pu[1], Wenhao Ren[1], Changwei Shi[1], Jiasheng Meng[1], Bo Song[3] & Liqiang Mai[1]

The abundance of sodium resources indicates the potential of sodium-ion batteries as emerging energy storage devices. However, the practical application of sodium-ion batteries is hindered by the limited electrochemical performance of electrode materials, especially at the anode side. Here, we identify alkaline earth metal vanadates as promising anodes for sodium-ion batteries. The prepared calcium vanadate nanowires possess intrinsically high electronic conductivity ($>100\,S\,cm^{-1}$), small volume change ($<10\%$), and a self-preserving effect, which results in a superior cycling and rate performance and an applicable reversible capacity ($>300\,mAh\,g^{-1}$), with an average voltage of $\sim 1.0\,V$. The specific sodium-storage mechanism, beyond the conventional intercalation or conversion reaction, is demonstrated through in situ and ex situ characterizations and theoretical calculations. This work explores alkaline earth metal vanadates for sodium-ion battery anodes and may open a direction for energy storage.

[1] State Key Laboratory of Advanced Technology for Materials Synthesis and Processing, Wuhan University of Technology, Wuhan 430070, China. [2] College of Physics and Electronic Engineering, Sichuan Normal University, Chengdu 610101, China. [3] Terahertz Technology Innovation Research Institute, School of Optical-Electrical Computer Engineering, University of Shanghai for Science and Technology, Shanghai 200093, China. Correspondence and requests for materials should be addressed to C.N. (email: niuchaojiang11@whut.edu.cn) or to B.S. (email: bsong@usst.edu.cn) or to L.M. (email: mlq518@whut.edu.cn)

The development of renewable energy, such as solar, wind, and tidal energy, is fundamentally important for our society due to the increasing scarcity of fossil fuels[1]. Large-scale energy storage systems are necessary to meet the challenge of discontinuity of the renewable energy flow[2, 3]. Recently, sodium-ion batteries (SIBs) have been considered as a promising battery technology and have attracted great attention[4]. SIBs have a similar configuration and electrochemical reaction processes with lithium-ion batteries (LIBs). But sodium resources are much more abundant and cost-effective than lithium resources, which makes SIBs more suitable for large-scale energy storage applications[5–7]. However, the performance of SIBs cannot currently meet the demands. The problem may be due to the larger ionic size of $Na^+$ or generally weak binding of Na with substrate, which results in sluggish reaction kinetics or severe degradation of the electrodes[8–10]. One of the most important tasks to realize the practical application of SIBs is to find suitable electrode materials with long-term cycling stability, high rate capability, low cost and high capacity.

Due to the unremitting efforts of scientists around the world, cathode research has made great progress[11–17]. However, the development of anodes faces a significant challenge[18]. Carbon-based materials (such as hard carbon, Fig. 1a) are limited by a poor rate capability and safety issues[19–21]. Titanium-based anodes (such as $NaTi_2(PO_4)_3$ and $Na_2Ti_3O_7$, Fig. 1a) store Na ions through an intercalation/de-intercalation mechanism, leading to low capacity due to limited intercalation sites[22–27]. Other anode materials based on conversion or alloying reactions (such as $SnO_2$, Sb and P, Fig. 1a) can deliver high initial capacity[28–34], but the inevitable large volume variation (Fig. 1b) during the discharge/charge process leads to the pulverization of the electrodes and subsequent severe capacity fading[35, 36]. Carbon modification and complex structure design can improve the electrochemical performance, but inevitably increase the cost and are not beneficial to practical applications. In this case, new anode materials with intrinsically good electrochemical properties are desired for the development of SIBs.

Vanadium-based materials, including vanadium oxides[37–42], alkaline metal vanadates[43], and transition metal vanadates[44–46], have been widely studied as electrodes of rechargeable batteries for more than 30 years[47–49]. However, the investigation of vanadium-based materials has been ignored in the context of SIB anodes. Vanadium can achieve multi-electron transfer below 1.0 V due to its multivalent properties[41], indicating that it may deliver higher capacity than Ti-based anodes. Additionally, the valence state of vanadium is not able to reach zero to form metallic V during the discharge/charge process due to the strong V–O bond strength[42], suggesting that the volume variation of vanadium-based electrodes at low voltage may be smaller than that of typical conversion-based electrodes (such as $SnO_2$ and $Fe_2O_3$). Moreover, previous studies showed that alkaline-earth-metal-containing compounds, especially Ca-containing compounds, such as Ca–Co–O[50], Ca–Sn–O[51], and Ca–Fe–O[52], exhibit interesting electrochemical performance when used as LIB anodes. Ca ions are electrochemically inactive, but can form nano-sized CaO and cause a "spectator effect" to buffer the volume change of the electrodes and to restrain the agglomeration of active nanograins[52]. Because of the "spectator effect" of nano-sized alkaline earth oxides (such as CaO) and the unique properties of vanadium (multivalent and strong V–O bond strength), alkaline earth metal vanadates are expected to exhibit both high capacity and long-term cycling stability as SIB anodes. And the unique advantages of nanowires in energy storage[53] and in situ characterization[54] are our motivation to fabricate nanowires.

Herein, we demonstrate the outstanding electrochemical properties of alkaline earth metal vanadates (Ca–V–O and Sr–V–O) as SIB anodes. Based on a single nanowire device, $CaV_4O_9$ shows good electronic conductivity. Through in situ/ex situ transmission electron microscopy (TEM) and X-ray diffraction (XRD) measurements together with ab initio calculations, a specific $Na^+$ storage mechanism, beyond the conventional intercalation or conversion reaction, is demonstrated. Ascribe to that, the electrodes can achieve four-electron transfer per formula

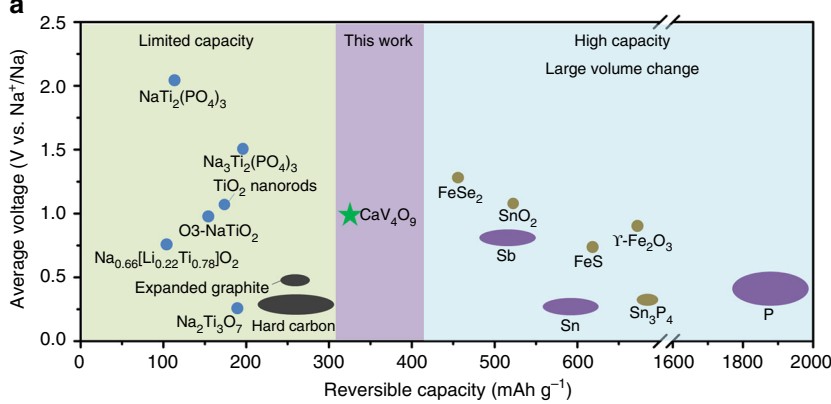

**Fig. 1** Comparison of the $CaV_4O_9$ nanowires in this work and previously reported SIB anodes. **a** Average voltage vs. reversible capacity of the anodes for SIBs. **b** Volume change of the reported alloying or conversion reaction anodes for SIBs

| Material | Volume change (%) | Material | Volume change (%) |
|---|---|---|---|
| $CaV_4O_9$ (this work) | < 10 | $Sn_3P_4$ | > 300 |
| P | ~ 500 | $SnO_2$ | > 300 |
| Sn | ~ 500 | $Fe_2O_3$ | ~ 215 |
| Sb | ~ 390 | FeS | ~ 170 |

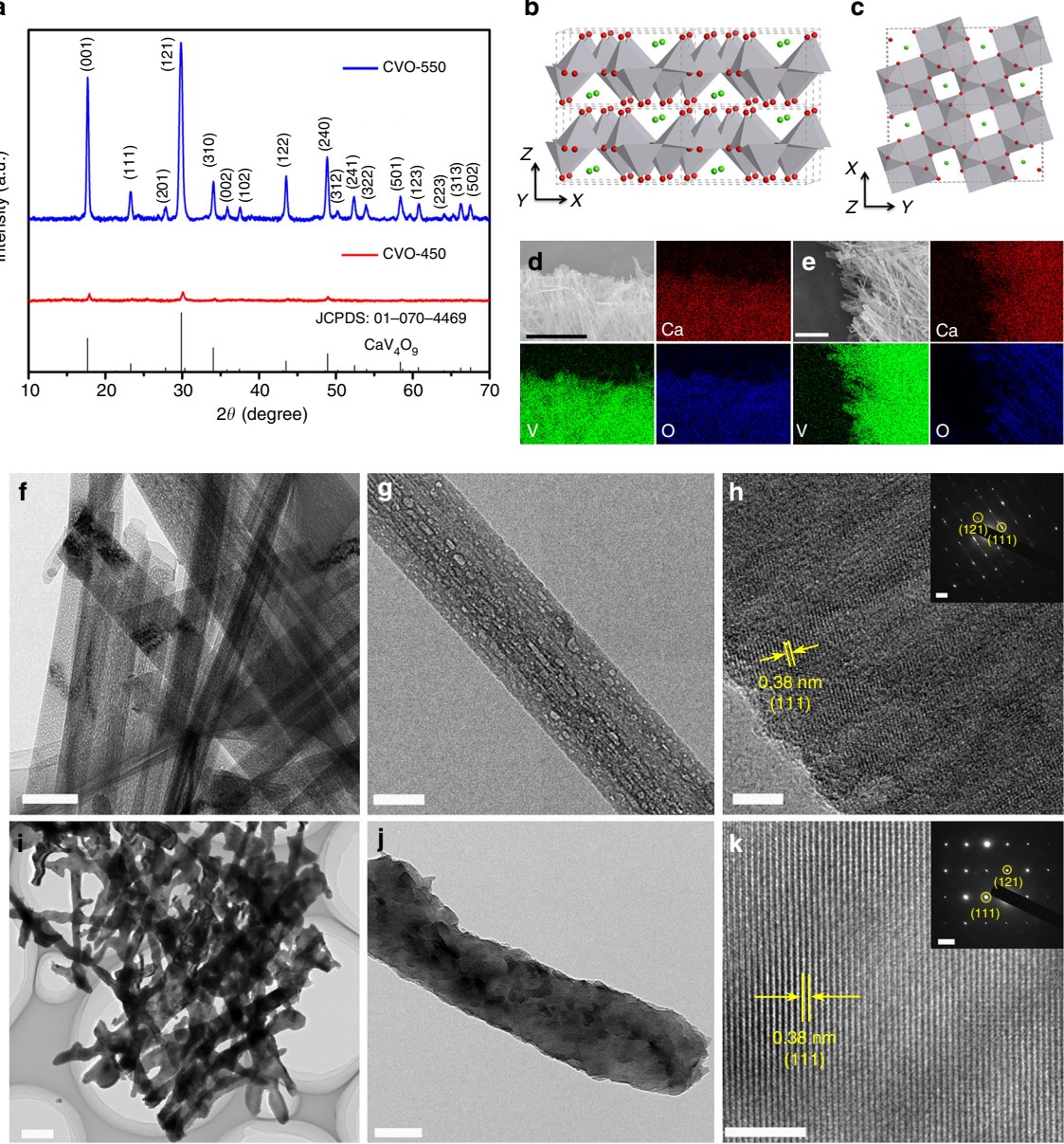

**Fig. 2** Characterization of CVO-450 and CVO-550 nanowires. **a** XRD patterns of CVO-450 and CVO-550. **b, c** Crystal structures of the CaV$_4$O$_9$; the *green* and *red balls* represent Ca and O ions, respectively, and the *grey polyhedrals* represent the V–O pyramids. **d, e** EDS mapping of CVO-450 and CVO-550, respectively. The *scale bar* is 6 μm for **d** and 3 μm for **e**. **f–h** TEM and HRTEM images of CVO-450; the inset of **h** is the SAED pattern of CVO-450. The *scale bars* of **f–h** and the inset of **h** are 200 nm, 50 nm, 5 nm, and 2 nm$^{-1}$, respectively. **i–k** TEM and HRTEM images of CVO-550; the inset of **k** is the SAED pattern of CVO-550. The *scale bars* of **i–k** and the inset of **k** are 500 nm, 100 nm, 5 nm, and 2 nm$^{-1}$, respectively

with negligible volume variation during the sodiation/desodiation process; this process is accompanied by the self-preserving effect from in situ formed CaO nanograins. These positive properties of CaV$_4$O$_9$ result in its applicable capacity, good rate capability, and long-term cycling stability, manifesting great potential in SIBs.

## Results

**Characterization of CaV$_4$O$_9$ nanowires.** CaV$_4$O$_9$ nanowires, with a diameter of ~100 nm and a length of ~10 μm (Supplementary Fig. 1), were synthesized using a facile hydrothermal method followed by heat treatment. Both low-crystalline and high-crystalline CaV$_4$O$_9$ nanowires were obtained by controlling the sintering temperature (Fig. 2a). The XRD results show that the sample prepared at 450 °C (marked as CVO-450) displays much weaker diffraction peaks than the sample prepared

at 550 °C (marked as CVO-550), indicating the low crystallinity of CVO-450 (Fig. 2a). The diffraction peaks of both CVO-450 and CVO-550 can be indexed to the pure phase of tetragonal CaV$_4$O$_9$ (JCPDS: 01-070-4469), which is a layered structure, and the Ca ions are evenly distributed in the crystal structure (Fig. 2b, c). Inductively coupled plasma (ICP) measurements indicate Ca/V molar ratios of 0.97/4 and 1.08/4 for CVO-450 and CVO-550, respectively. Energy-dispersive X-ray spectra (EDS) mappings demonstrate that Ca, V, and O uniformly exist in both samples (Fig. 2d, e). For comparison, we also prepared V–O nanowires (Supplementary Fig. 2) using a similar process without adding Ca. The nanowires were characterized as the monoclinic VO$_2$ phase (JCPDS: 01-076-0456) (marked as VO$_2$-450).

TEM and high-resolution TEM (HRTEM) images of CVO-450 are shown in Fig. 2f–h; large amounts of cavities are distributed on the nanowires (Fig. 2g and Supplementary Fig. 3). However,

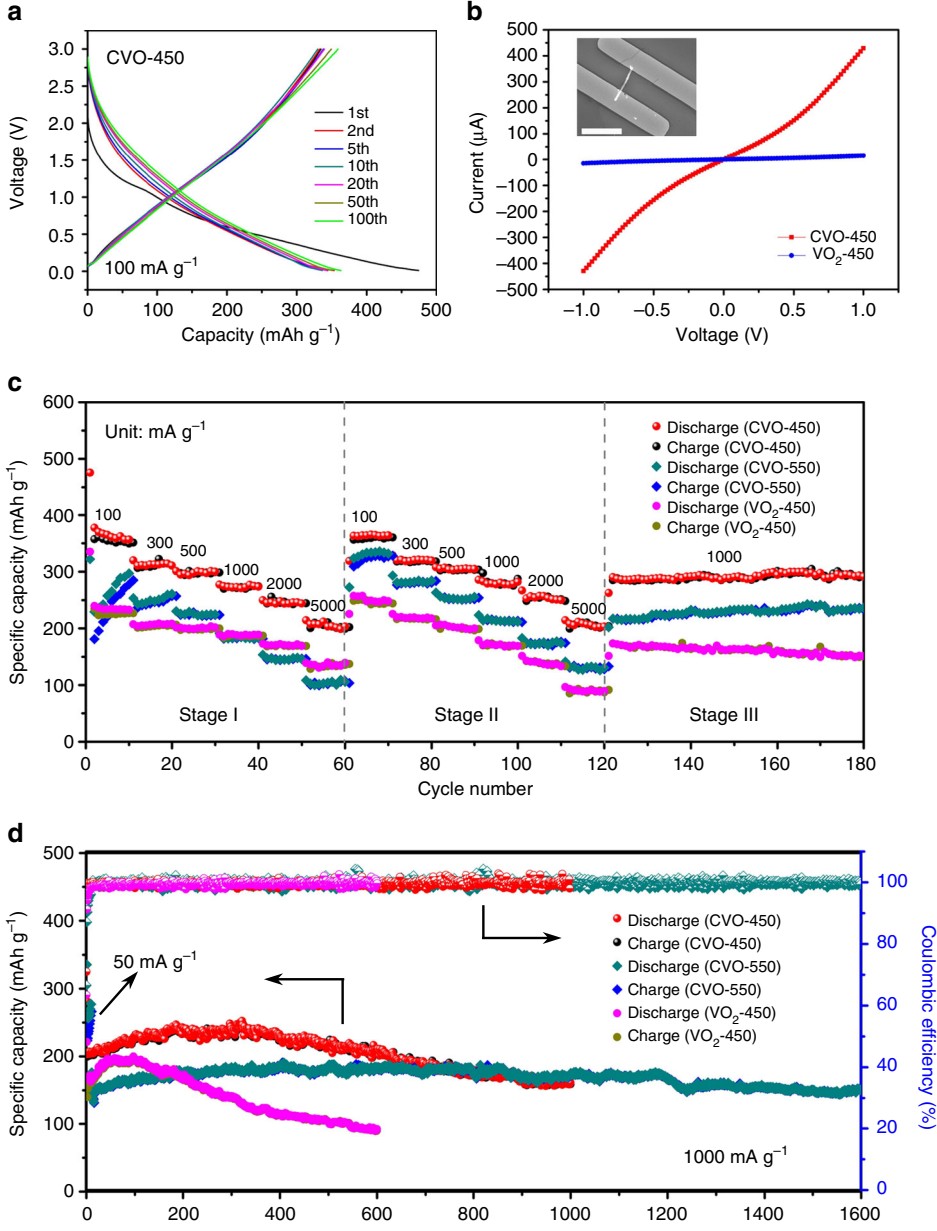

**Fig. 3** Electrochemical properties of CaV$_4$O$_9$ nanowires and VO$_2$ nanowires as SIB anodes. **a** Discharge/charge profiles of CVO-450 after different numbers of cycles at 100 mA g$^{-1}$. **b** I–V curves of CVO-450 and VO$_2$-450 nanowires. Inset is the SEM image of the CVO-450 single nanowire device. *Scale bar*, 10 μm. **c** Rate performances of CVO-450, CVO-550, and VO$_2$-450 at different current densities of 100, 300, 500, 1000, 2000 and 5000 mA g$^{-1}$. **d** Cycling performances and Coulombic efficiency of CVO-450, CVO-550, and VO$_2$-450 at a current density of 1000 mA g$^{-1}$

no obvious cavities were observed for CVO-550 (Fig. 2i, j), and the morphology is not as regular as that of CVO-450 because of the higher annealing temperature. The HRTEM image of CVO-450 shows distinct amorphous regions with lattice fringes (Fig. 2h), indicating the low crystallinity. However, the highly crystalline CVO-550 shows a defined lattice fringe with inter-planar spacing of 0.38 nm, corresponding to the (111) lattice plane, and the selected area electron diffraction (SAED) pattern reveals the single-crystal characteristic of the CVO-550 nanowires (Fig. 2k).

Thermogravimetry/derivative thermogravimetry (TG/DTG), Fourier-transformed infrared (FT-IR), and XRD measurements were performed to explore the formation mechanism of the low-crystalline structure with large amounts of cavities of the CVO-450 nanowires (Supplementary Figs. 4, 5). These results demonstrated that the evaporation of crystal water leads to the formation of the low-crystalline structure and the cavities on the CVO-450 nanowires (Supplementary Fig. 6, Supplementary Note 1). While for CVO-550, the higher annealing temperature leads to the higher crystallinity and the closure of the cavities together with the irregular morphology. Details can be found in Supplementary Information.

**Electrochemical properties of CaV$_4$O$_9$ nanowires.** We explored and compared the electrochemical performance of the two calcium vanadate nanowire samples (CVO-450 and CVO-550) as SIB anodes to evaluate the potential of CaV$_4$O$_9$ as an electrode material. Cyclic voltammetry (CV) results were measured at 0.1 mV s$^{-1}$ in the voltage range of 3.0–0.01 V (vs. Na$^+$/Na). CVO-450 shows no obvious redox peaks, but CVO-550 displays

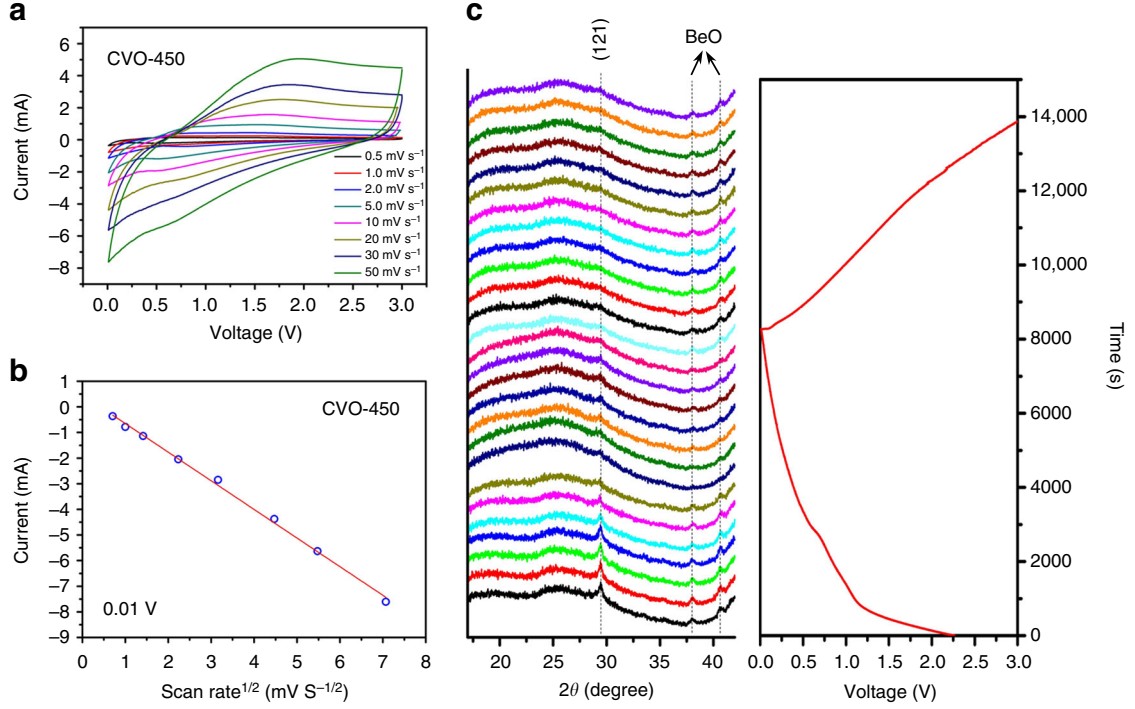

**Fig. 4** Cyclic voltammetry and in situ XRD results of CVO-450. **a** CV curves of CVO-450 at scan rates from 0.5 to 50 mV s$^{-1}$. **b** The relation between the square root of the scan rate ($v^{1/2}$) and the corresponding currents at 0.01 V. **c** In situ XRD results of CVO-450 during the initial discharge and charge process

an irreversible cathodic peak at ~1.1 V in the initial scan (Supplementary Fig. 7a, b), which may correspond to the irreversible structural dissociation in the initial sodiation process. The capacity loss below 0.5 V for both CVO-450 and CVO-550 in the initial scan can be attributed to the formation of a solid electrolyte interphase and unmanageable side reaction at low voltage[55, 56]. The discharge/charge profiles (Fig. 3a) at a current density of 100 mA g$^{-1}$ (voltage range from 3.0 to 0.01 V vs. Na$^+$/Na) show that CVO-450 exhibits a reversible capacity of ~350 mAh g$^{-1}$ with an initial Coulombic efficiency of 70.5%. There is no distinct voltage plateau in the charge/discharge curves, with an average voltage of ~1.0 V, in agreement with the CV results. The voltage hysteresis between discharge and charge, which may be attributed to the different reaction paths during sodiation and desodiation[57], shows a minor change from the 2nd cycle to the 100th cycle, indicating a good electrochemical stability. The capacity retention is 102.8% relative to the second discharge capacity after 100 cycles (Supplementary Fig. 7e), demonstrating a good cycling stability. CVO-550 exhibits a distinct capacity increase in the initial 20 cycles with lower initial Coulombic efficiency (Supplementary Fig. 7f), and the corresponding discharge/charge curves display reduced polarization, indicating activation and amorphization. The electrochemical impedance spectroscopy (EIS) results (Supplementary Fig. 8) show that the charge transfer resistance ($R_{ct}$) of CVO-450 after the first cycle is much smaller than that of CVO-550. After different cycles, the $R_{ct}$ value for CVO-450 shows only minor changes, which is consistent with its stable cycling performance. While for CVO-550, $R_{ct}$ decreases significantly after 100 cycles, confirming the activation process during cycling.

To investigate the superiority of CaV$_4$O$_9$ compared to pristine vanadium oxide with regard to their electrochemical properties, we compared the electric conductivity of CVO-450 and VO$_2$-450 based on the $I$–$V$ curves by assembling single nanowire devices (Fig. 3b). The conductivity was 110–265 S cm$^{-1}$ for CVO-450,

~15 times higher than that of VO$_2$-450 (7–14 S cm$^{-1}$). Note that the conductivity of the low-crystallinity CaV$_4$O$_9$ nanowire exceeds 100 S cm$^{-1}$ and is higher than that of VO$_2$ nanowires. The improved conductivity may be attributed to the increased carrier density introduced by Ca ions in the structure[58].

The rate capabilities of CVO-450, CVO-550, and VO$_2$-450 were tested to compare their electrochemical performance (Fig. 3c). All three electrodes go through three stages at different current densities. CVO-450 has an average capacity of 363.9 mAh g$^{-1}$ at 100 mA g$^{-1}$ in stage I. When the current rate increased to 5000 mA g$^{-1}$ (25 C), the average capacity still reached 205.0 mAh g$^{-1}$ (Fig. 3c and Supplementary Fig. 9), which is 56.3% of that at 100 mA g$^{-1}$, despite the 50-fold increase in the current density, indicating outstanding rate capability. Notably, when the current density returned to 100 mA g$^{-1}$ and then increased to 5000 mA g$^{-1}$ in stage II, no capacity decay was observed but there was a slight capacity increase compared to the corresponding rate in stage I (Supplementary Table 1). In stage III, CVO-450 still displays a slight increasing trend in capacity at 1000 mA g$^{-1}$. CVO-550 displays lower capacity and inferior high rate capability compared with CVO-450. However, its increasing trend is more distinct among the three stages. The average capacity at 100 mA g$^{-1}$ increased to 331.5 mAh g$^{-1}$ in stage II compared to stage I (265.2 mAh g$^{-1}$) (Fig. 3c, Supplementary Table 1). However, for VO$_2$-450, capacity fading is observed at high current density in stage II compared with that in stage I. And in stage III, the capacity fading is also distinct compared with the two CaV$_4$O$_9$ nanowire samples. These results indicate that CaV$_4$O$_9$ nanowires display better rate capability and electrochemical stability than VO$_2$ nanowires.

To further confirm the improved electrochemical stability of CaV$_4$O$_9$ nanowires over VO$_2$ nanowires, CVO-450, CVO-550, and VO$_2$-450 were tested at 1000 mA g$^{-1}$ for long-term cycling (Fig. 3d). CVO-450 exhibits a discharge capacity of 203.2 mAh g$^{-1}$ at the second cycle and a gradual increase in the

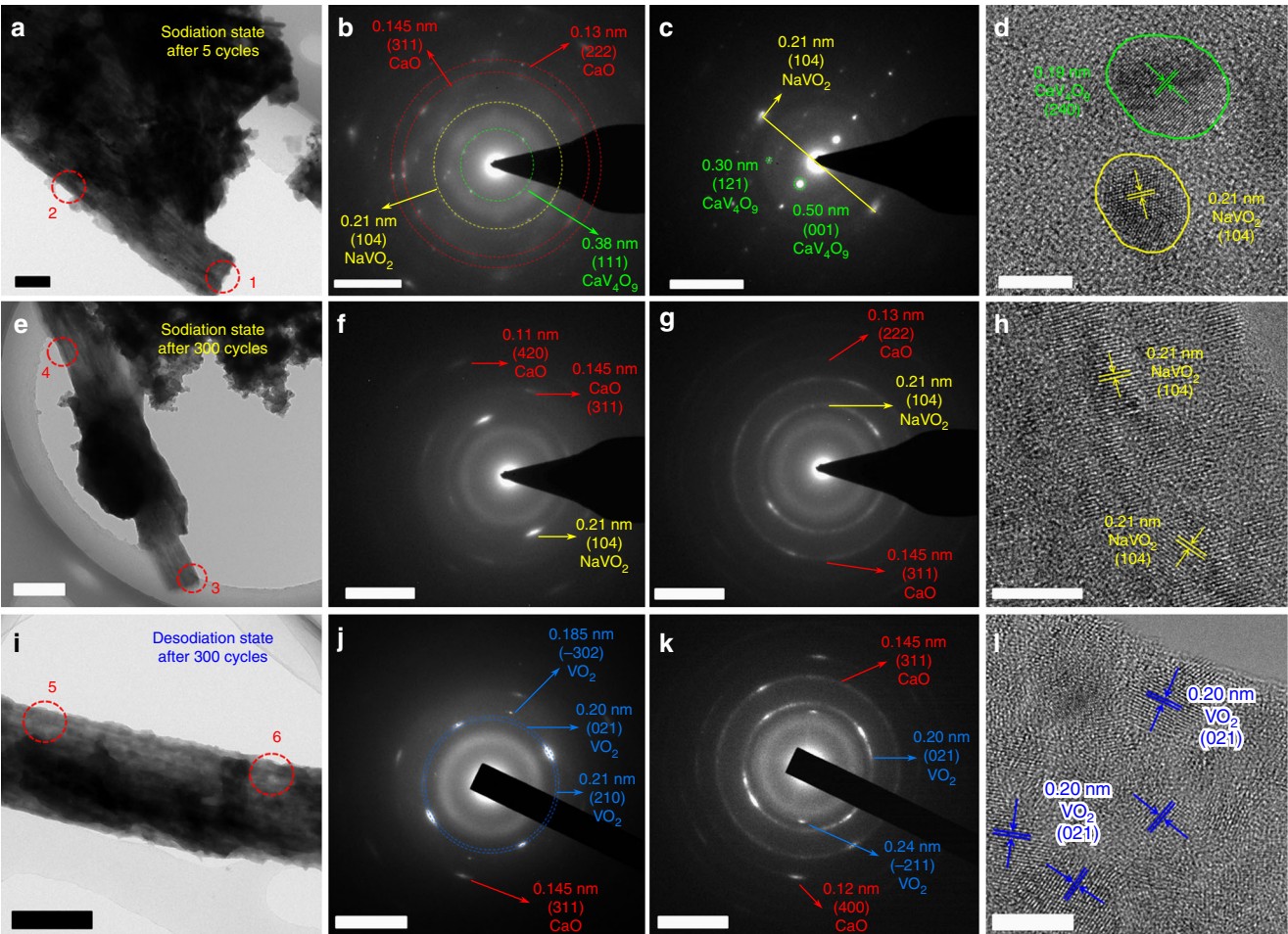

**Fig. 5** Ex situ TEM characterization of CVO-450 at the discharge and charge states. **a–d** TEM image, SAED patterns, and HRTEM image of CVO-450 at the sodiation state after 5 cycles at 1000 mA g⁻¹. The SAED patterns of **b** and **c** were collected from region 1 and region 2 in **a**, respectively. **e–h** TEM image, SAED patterns, and HRTEM image of CVO-450 at the sodiation state after 300 cycles at 1000 mA g⁻¹. The SAED patterns of **f** and **g** were collected from region 3 and region 4 in **e**, respectively. **i–l** TEM image, SAED patterns and HRTEM image of CVO-450 at the desodiation state after 300 cycles at 1000 mA g⁻¹. SAED patterns of **j** and **k** were collected from region 5 and region 6 in **i**, respectively. *Scale bars*: **a** 200 nm; **e**, **i** 500 nm; **b**, **c**, **f**, **g**, **j**, **k** 5 nm⁻¹; **d**, **h**, **l** 5 nm

following ~300 cycles. After 1000 cycles, the discharge capacity remains at 161.5 mAh g⁻¹, corresponding to a retention of 79.4% relative to the second discharge. For CVO-550, the electrode was activated at 50 mA g⁻¹ for 10 cycles and then cycled at 1000 mA g⁻¹. The 12th discharge capacity is 146.2 mAh g⁻¹, and after 1600 cycles, the capacity remains at 149.8 mAh g⁻¹, corresponding to a capacity retention of 102.4% and demonstrating excellent cycling stability. Note that the CVO-550 shows better cycling stability than CVO-450. For VO₂-450, the second discharge capacity is 166.9 mAh g⁻¹, and its capacity also increases at first and then decreases. After 600 cycles, the capacity decreases to 90.3 mA g⁻¹, with a capacity retention of 54.1% (Fig. 3d). The long-term cycling performance also demonstrates that CaV₄O₉ nanowires possess better cycling stability than VO₂ nanowires.

Na-ion full batteries with CVO-450 as the anode and excessive Na₃V₂(PO₄)₃/C nanoparticles (Supplementary Fig. 10) as the cathode were assembled, to demonstrate the potential of the as-synthesized calcium vanadate nanowires for practical applications. Na₃V₂(PO₄)₃ was selected as the counterpart due to its stable voltage plateau and excellent structural stability when used as a SIB cathode. The full cell, cycled in the voltage range of 0.5–3.2 V, exhibits an initial discharge capacity of 165.6 mAh g⁻¹ at 1000 mA g⁻¹ based on the mass of CVO-450. The charge/discharge profiles display sloping curves, with an average output voltage of ~1.8 V (Supplementary Fig. 11b). This feature of the sloping charge/discharge curve is beneficial for monitoring the charge/discharge state of the batteries, which is significant for practical applications. After 400 cycles, the discharge capacity is maintained at 170.1 mAh g⁻¹ (Supplementary Fig. 11c), indicating good cycling stability. Full batteries with higher output voltage and smaller voltage hysteresis can be produced by using cathodes with higher voltages and optimizing the transport properties of both electrodes.

**Sodium-storage mechanism**. As mentioned above, except the initial discharge process, both CVO-450 and CVO-550 display sloping discharge/charge profiles without voltage plateaus. To clarify the Na⁺ storage mechanism, CV measurements for CVO-450 (Fig. 4a) and CVO-550 (Supplementary Fig. 12a) were performed at different scan rates. The sodiation currents (at 0.01 V) display a linear relation with the square root of the scan rate ($v^{1/2}$) for both samples (Fig. 4b and Supplementary Fig. 12b), which indicates a diffusion-controlled charge storage mechanism for CVO-450 and CVO-550 but not a capacitive process[59, 60]. To obtain deeper insight, in situ XRD measurement was performed on CVO-450 (Fig. 4c). In the initial sodiation

process, no shift was observed for the (121) peak at about 29.8°, but the peak intensity weakened as the discharge proceeded, indicating the consumption of pristine $CaV_4O_9$ and the amorphization of the structure. At the end of the discharge, the (121) peak completely disappeared and did not return during the charge process. Ex situ XRD tests were also performed on CVO-550. None of the peaks shifted, but they only weakened (Supplementary Fig. 13), displaying the same trends as the in situ XRD characterization. These results suggest that Na ions are stored by reacting with $CaV_4O_9$ directly, and this process leads to amorphization of the pristine structure without bulk phase transformation, which accounts for the sloping discharge/charge curves. Moreover, the amorphous or less-crystalline structure provides more active sites and better ion diffusion kinetics[42, 46], which explains the significant increase in capacity for CVO-550 in the initial cycles. And due to the low crystallinity of CVO-450 in the pristine state, it displays a better rate capability than CVO-550.

The valence variation during the $Na^+$ insertion/extraction was investigated by *ex situ* X-ray photoelectron spectroscopy (XPS) measurement on CVO-450 (Supplementary Fig. 14). The peak at the binding energy of 347.5 eV for the electrode before cycling is attributed to the $Ca^{2+}$ 2p3/2, and the peak of V 2p3/2 (516.5 eV) corresponds to $V^{4+}$. No distinct chemical shift of the binding energy for the $Ca^{2+}$ 2p3/2 was observed for the electrode after initial discharge to 0.01 V, revealing the inactivity of $Ca^{2+}$. For the V 2p3/2 spectrum, the peak can be fitted to two portions. The new peak at 515.4 eV indicates the formation of $V^{3+}$, whereas the binding energy at 516.8 eV indicates the coexistence of $V^{4+}$ in the discharge state, suggesting the incomplete reaction of $CaV_4O_9$ in the initial discharge. The valance of V is mainly recovered to +4 after charging to 3.0 V, while the Ca is unchanged.

Ex situ TEM measurements were performed on CVO-450 to study the structural change during $Na^+$ insertion/extraction. The nanowire morphology remains integrated at the sodiation state after five cycles (Fig. 5a). The SAED patterns collected from region 1 and region 2 are presented in Fig. 5b, c, respectively. The diffraction spots in the *green circle* (Fig. 5b) can be indexed to the (111) plane of $CaV_4O_9$ (Supplementary Table 2), which indicates the existence of unreacted $CaV_4O_9$. The diffraction spots in the *yellow circle* correspond to lattice planes with a spacing of 0.21 nm. The XPS results indicate that $Na^+$ insertion leads to the formation of $V^{3+}$. Combined with the XPS and SAED results, the diffraction spots with a spacing of 0.21 nm can be regarded as the (104) lattice plane of $NaVO_2$ (JCPDS: 00-027-0825, Supplementary Table 3). The formation of $NaVO_2$ indicates the decomposition of pristine $CaV_4O_9$, and CaO nanograins will be formed. The rest of the diffraction spots in the *red circles* correspond to spacings of 0.145 and 0.13 nm, confirming the generation of CaO (JCPDS: 00-001-1160, Supplementary Table 4). In the diffraction pattern of region 2 (Fig. 5c), more obvious diffraction spots from $CaV_4O_9$ and diffraction ring from $NaVO_2$ were detected, further confirming the coexistence of $CaV_4O_9$ and $NaVO_2$ at the sodiation state. The HRTEM image (Fig. 5d) shows nanograins with size of ~5 nm, and lattice fringes with spacings of 0.19 and 0.21 nm, which correspond to the (240) plane of $CaV_4O_9$ and the (104) plane of $NaVO_2$, respectively. The HRTEM image is consistent with the SAED patterns and demonstrates that $Na^+$ insertion in the initial cycles leads to partial conversion from $CaV_4O_9$ to $NaVO_2$ and CaO nanograins.

For CVO-450 at the sodiation state, even after 300 cycles, the nanowire retains its integrity (Fig. 5e). Both the SAED patterns collected from regions 3 and 4 display a distinct diffusion ring (Fig. 5f, g), indicating the formation of amorphous phase in the nanowires. Moreover, the typical diffraction rings of $NaVO_2$ and CaO were also detected (Fig. 5f, g). But no diffraction spots or

rings of $CaV_4O_9$ were found, revealing the complete reaction of the original $CaV_4O_9$. These results indicate that the unreacted $CaV_4O_9$ gradually participates in the reaction, as demonstrated by the observed capacity increase of CVO-450 in the first ~300 cycles (Fig. 3d). The HRTEM image shows the (104) plane of $NaVO_2$ with different orientations (Fig. 5h), consistent with the SAED patterns. Note that no obvious lattice fringes of CaO were observed in the HRTEM images, suggesting the small size of the CaO nanograins in the matrix.

For the desodiation state of the CVO-450 after 300 cycles, SAED patterns were also collected from two regions of the nanowires (Fig. 5i). The diffraction rings of the (311) plane (Fig. 5j, k) and (400) plane (Fig. 5k) of CaO were also detected, further confirming the "spectator effect" of the Na-driven CaO. Moreover, in Fig. 5j, a much thicker diffraction ring in the *blue circles* was observed, which suggests the overlap of two or three rings. In addition, the inner and outer rings correspond to plane spacings of 0.21 and 0.20 nm, respectively, which is consistent with the (210) and (021) planes of monoclinic $VO_2$ (JCPDS: 00-009-0142, Supplementary Table 5). A weak diffraction ring with a spacing of 0.185 nm was also detected, consistent with the (−302) plane of $VO_2$ (Supplementary Table 5). The diffraction rings with plane spacings of 0.24 and 0.20 nm in Fig. 5k further confirm the formation of $VO_2$ at the desodiation state. The HRTEM image shows obvious lattice fringes with a spacing of 0.20 nm, corresponding to the (021) plane of $VO_2$, consistent with the SAED patterns. The SAED and HRTEM results indicate the conversion of $NaVO_2$ nanograins to $VO_2$ nanograins in the $Na^+$ extraction process. Thus, the total reaction can be summarized as follows:

The initial discharge process:

$$CaV_4O_9 + 4Na^+ + 4e^- \rightarrow 4NaVO_2 + CaO \qquad (1)$$

The subsequent cycles:

$$NaVO_2 \leftrightarrow VO_2 + Na^+ + e^- \qquad (2)$$

There is no metallic V or $Na_2O$ generated both in the initial discharge process and the subsequent cycles, which is different from the typical conversion reaction observed in other transition metal oxides.

The ex situ TEM measurement was also performed on $VO_2$ nanowires at the sodiation state after 300 cycles. The nanowire morphology also keeps integrated according to the TEM image (Supplementary Fig. 15a). The SAED pattern and the HRTEM image manifest the formation of $NaVO_2$ (JCPDS: 00-044-0342) at the sodiation state (Supplementary Fig. 15b, c). However, different from that observed at the sodiation state of $CaV_4O_9$, the SAED pattern displays strong diffraction spots rather than diffraction rings (Supplementary Fig. 15b), indicating that the generated $NaVO_2$ display a large grain size. From the HRTEM image, the distinct nanograins were observed with grain size of ~20 nm (Supplementary Fig. 15c), much larger than those derived from $CaV_4O_9$.

To confirm the proposed reaction mechanism of $CaV_4O_9$, we performed ab initio calculations based on the density-functional theory (DFT). The difference of total energies between the resultants and reactants in Eq. (1) reaches −6.13 eV, indicating that the resultants are much more stable than the reactants. Therefore the conversion from $CaV_4O_9$ to $NaVO_2$ is spontaneous after the $Na^+$ insertion and not reversible. Next, considering the small size of generated $NaVO_2$ and $VO_2$ nanograins (~5 nm, observed in TEM characterization), we used three models of (001), (010), and (110) surfaces (Supplementary Fig. 16a–c) to study the surface effect of $NaVO_2$ nanograins on the Na dissociation. The calculated dissociation energies of Na from the

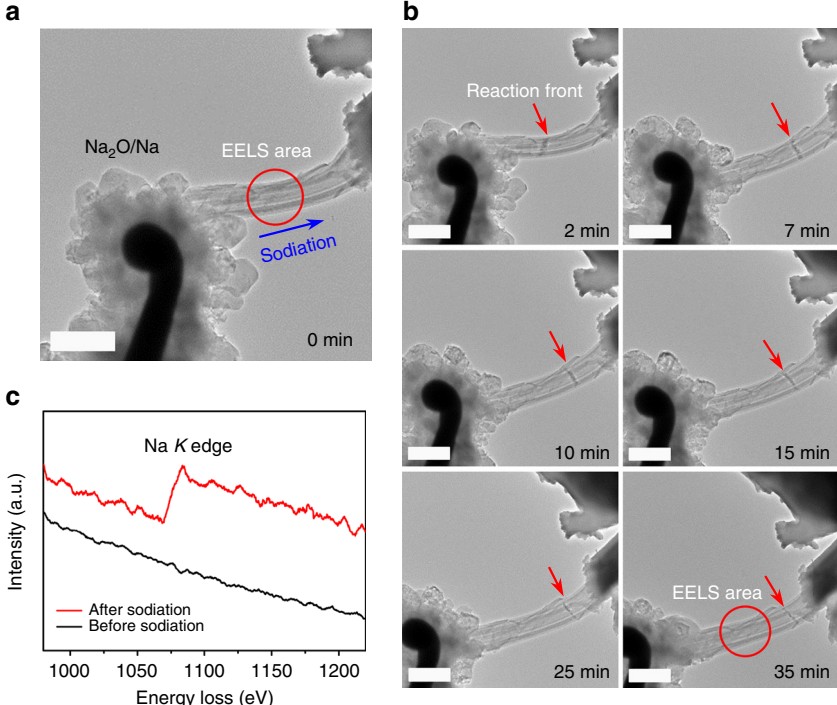

**Fig. 6** In situ TEM results of CVO-450. **a** Configuration of the in situ TEM device. *Scale bar*, 500 nm. **b** Time-lapse TEM images after applying a voltage bias. *Scale bar*, 500 nm. **c** EELS data collected from the *red circle* area at 0 and 35 min

(001), (010), and (110) surfaces were −0.23, 0.50, and 0.41 eV, respectively. These small values indicate that the Na dissociation from all the three surfaces is feasible and highly reversible, suggesting a reversible conversion between $NaVO_2$ and $VO_2$ nanograins as Eq. (2) in charge/discharge processes. Moreover, a model of $NaVO_2$ crystal (Supplementary Fig. 16d) was also applied to calculate the dissociation energy of Na. In this case, the average energy reached a large value of 5.63 eV, indicating the conversion from $NaVO_2$ crystal to $VO_2$ is hard to be realized. Thus, the reaction of Eq. (2) is influenced by the particle size, and the small generated nanograins (~5 nm) derived from $CaV_4O_9$ are beneficial to the conversion between $NaVO_2$ and $VO_2$. Besides, these results highlight the importance of the generated CaO, which prevent the $NaVO_2$ nanograins from agglomerating and preserve the small size of the active components, and then keep the electrodes with high reversibility.

## Discussion
Based on Eqs. (1) and (2), the theoretical specific capacity and volume change of $CaV_4O_9$ can be calculated. Assuming that $CaV_4O_9$ is fully reacted in the initial sodiation process, as shown in Eq. (1), four electrons are transferred when $V^{4+}$ is converted to $V^{3+}$, and the corresponding capacity is 276.4 mAh g$^{-1}$. Note that the average capacity of CVO-550 at 100 mA g$^{-1}$ reaches 331.5 mAh g$^{-1}$ during stage II (Fig. 3c), which is higher than the calculated value. This is because Na$^+$ insertion also leads to the formation of an amorphous phase (as evidenced by the ex situ TEM results) in addition to $NaVO_2$ and CaO nanograins, and the amorphous region provides additional active sites (such as vacant sites or void spaces) for Na ions[42, 61], resulting in the higher capacity. The capacity of CVO-450 is higher than that of CVO-550 at different current densities. The extra capacity can be attributed to the residual water in the nanowires. As described before, the nanowire samples contain considerable crystal water (~6.2%) before annealing (Supplementary Figs. 4–6). As indicated by the XRD and FT-IR results (Supplementary Fig. 5a, c), the

crystal water was still not fully removed for CVO-400 (annealed at 400 °C); thus, it is reasonable that there is minor residual water in CVO-450. As reported by Grey's group[56], residual water or OH groups can act as a major source of additional capacity. However, residual water also leads to the consumption and decomposition of the electrolyte and does not benefit the long-term cycle life of the batteries, which explains why CVO-450 has inferior cycling stability compared to CVO-550 (Fig. 3d). To further confirm this point, CVO-400 was also tested as electrode and compared with CVO-450 and CVO-550. As expected, the CVO-400 exhibits higher initial capacity but much poorer cycling stability (Supplementary Fig. 17).

For the volume change, we also assumed the complete reaction of $CaV_4O_9$ in the initial cycle. Based on Eq. (1), one mole of $CaV_4O_9$ is converted to four moles of $NaVO_2$ and one mole of CaO. The cell volumes of $CaV_4O_9$, $NaVO_2$, and CaO are 347.6 Å$^3$ ($Z = 2$), 126.7 Å$^3$ ($Z = 4$), and 110.4 Å$^3$ ($Z = 4$) ($Z$ represents the number of molecules in each unit cell), respectively. The volume change of $CaV_4O_9$ in the sodiation process is calculated as follows:

$$\text{Volume change} = \frac{4 \times \frac{1}{4} \times 126.7 + \frac{1}{4} \times 110.4 - \frac{1}{2} \times 347.6}{\frac{1}{2} \times 347.6} = -11.2\%$$

For the desodiation process, one mole of $NaVO_2$ is converted to one mole of $VO_2$, as shown in Eq. (2). The cell volume of $VO_2$ is 117.5 Å$^3$ ($Z = 4$); thus, the volume change is –7.3%. Notably, the theoretical calculation indicates tiny volume shrinkage in the initial sodiation/desodiation process. However, because of the stepwise reaction of $CaV_4O_9$ in the initial cycles and the fact that the amorphous phase (which was not taken into account during the calculation) is also formed during the discharge/charge process, the shrinkage is effectively offset. Thus, the volume change in the entire discharge/charge process is very small (<10% based on a conservative estimation), which explains why the nanowire morphology is retained in both the discharge and charge states, even after 300 cycles. The small volume change

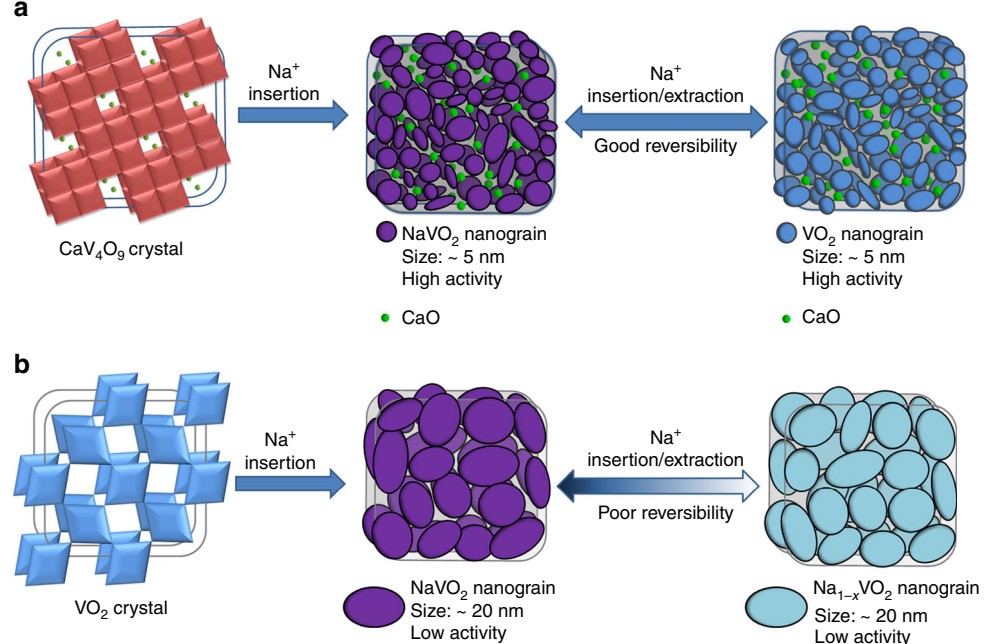

**Fig. 7** Illustration of the reaction mechanism during initial sodiation and subsequent cycles. **a** Sodium-storage mechanism of $CaV_4O_9$. **b** Sodium-storage mechanism of $VO_2$

during the sodiation/desodiation process essentially benefits the electrochemical stability[27].

In situ TEM was performed to confirm the small volume change of the $CaV_4O_9$ nanowires during the sodiation process. The in situ electrochemical device was assembled with CVO-450 nanowire as the working electrode and bulk metallic Na as the counter electrode. The naturally formed $Na_2O$ layer on the surface of metallic Na acted as the solid electrolyte[62] (Fig. 6a). The sodiation process was driven by an external constant voltage. The reaction front was clearly observed after the voltage was applied and diffused towards the other side of the nanowire (Fig. 6b). Electron energy loss spectroscopy (EELS) data (Fig. 6c) were collected from the *red circle* areas at 0 and 35 min, respectively, and confirmed the diffusion of $Na^+$ into the nanowire. Remarkably, during the entire sodiation process, the nanowire retains its integrity and remains steady, and there is no obvious expansion, elongation or bending, revealing the small volume change and good structural stability of $CaV_4O_9$ nanowires during the $Na^+$ insertion. The in situ TEM measurement further confirms the reliability of our analyses.

Based on our comprehensive analyses, the reaction mechanism of the $CaV_4O_9$ was proposed, as shown in Fig. 7a. The $Na^+$ insertion into $CaV_4O_9$ crystal results in the irreversible structural dissociation and amorphization, meanwhile, $NaVO_2$ and CaO nanograins are generated and dispersed uniformly in the amorphous matrix. In the desodiation process, the $NaVO_2$ nanograins are converted to $VO_2$ nanograins, and CaO act as a spectator. The superior electrochemical performance of $CaV_4O_9$ can be attributed to the following aspects: First, both the pristine $CaV_4O_9$ and the generated $VO_2$ have good electric conductivity, which ensures good conductance of the nanowires throughout the whole discharge/charge process. Second, the structural dissociation and amorphization process leads to increase of the active surface area, which is beneficial to the ion diffusion kinetics. The good conductance and the increased active surface area contribute to the good rate capability. Besides, the in situ generated and uniformly distributed CaO nanograins produce a self-preserving effect in the whole subsequent cycles, which

effectively inhibit the agglomeration of the active components, and then preserve the high reversibility of the conversion between $NaVO_2$ and $VO_2$ nanograins in the desodiation/sodiation process. Finally, the entire sodiation/desodiation process produces a small volume change of <10%, which further ensures the good cycling stability. In the case of $VO_2$ crystal (Fig. 7b), due to the absence of the self-preserving effect from CaO, the generated $NaVO_2$ nanograins tend to agglomerate into relatively large sized particles as the cycling continues, which results in the poor reversibility of the subsequent desodiation/sodiation processes, and then the observed capacity fading.

To reflect the generality of alkaline earth metal vanadates as high-performance SIB anodes, we fabricated Sr–V–O nanowires (Supplementary Fig. 18) through a similar synthesis method. The XRD results of samples annealed at different conditions are shown in Supplementary Fig. 18a. The TG curve shows a weight loss of 5.2% between 200 and 450 °C, corresponding to the evaporation of crystal water (Supplementary Fig. 18b). The sintered Sr–V–O nanowires did not form a specific crystalline phase after the crystal water was removed, which may be due to the higher formation energy of Sr–V–O crystals. The sample annealed at 450 °C (marked as SVO-450) was tested as an SIB anode. The electrochemical performance of SVO-450 is shown in Supplementary Fig. 19. Similar to Ca–V–O, SVO-450 shows sloping discharge/charge profiles. As expected, it displays both good rate capability and excellent cycling stability up to 2000 cycles.

In summary, we have shown that alkaline earth metal vanadates (Ca–V–O and Sr–V–O) are promising SIB anodes. The $CaV_4O_9$ nanowires possess a specific $Na^+$ storage mechanism beyond the typical intercalation or conversion reaction and exhibit multiple positive electrochemical properties, including good electric conductivity (>100 S cm$^{-1}$), four-electron transfer with a small volume change (< 10%) and a self-preserving effect (from in situ Na-driven CaO), which results in a reversible capacity over 300 mAh g$^{-1}$, a rate capability up to 5000 mA g$^{-1}$ (25 C) and excellent cycling stability that allows 1600 cycles. Moreover, the sloping discharge/charge profile, with an average

voltage of ~1.0 V, is beneficial to the safety of the batteries. Considering the exciting electrochemical properties and superior performance, together with the cost benefit of alkaline earth and vanadium resources, we believe these results will promote the development of SIBs. Moreover, our study also shows an unexploited field of alkaline earth metal vanadates for the exploration of electrode materials for Li/Na-ion batteries or even rechargeable Zn/Mg/Ca-ion batteries, and thus may open new directions in energy storages.

## Methods

**Materials synthesis.** In a typical synthesis, 2 mmol of $V_2O_5$ was dispersed into 30 ml of deionized water, and then 5 ml of $H_2O_2$ was added dropwise. After stirring for 20 min, a clear, transparent orange solution was obtained. Then, 90 mmol of $CaCl_2$ was added to the orange solution. After all the $CaCl_2$ was added, a drastic exothermic reaction was observed, and a dark red suspension was obtained. The suspension was stirred for an additional 2 h and was then transferred into a 50 ml Teflon-lined stainless steel autoclave. The autoclave was sealed and maintained at 200 °C for 4 days and was then cooled to room temperature. The orange products were dispersed in 30 ml of deionized water to form a homogeneous suspension, and washed with deionized water four times and rinsed with ethanol one time. After drying at 70 °C for 24 h, the sample was annealed in an $H_2$/Ar (5/95) atmosphere at a heating rate of 2 °C min$^{-1}$. The as-synthesized samples of CVO-450 and CVO-550 were annealed at 450 °C and 550 °C, respectively, for 8 h. For Sr–V–O, 20 ml of deionized water and 100 mmol of $SrCl_2 \cdot 6H_2O$ were used in the hydrothermal process, and the other processes were the same. For comparison, VO2-450 was synthesized without the addition of $CaCl_2$ or $SrCl_2$ in the hydrothermal process, and all other processes were the same.

**Characterization.** The crystal phases of the products were characterized by X-ray powder diffraction using a Bruker D8 Discover X-ray diffractometer equipped with a Cu Kα radiation source. Scanning electron microscopy images were collected using a JEOL-7100F microscope. EDS mapping was recorded using an Oxford EDS IE250 system. TEM, HRTEM and SAED were performed using a JEOL JEM-2100F STEM/EDS microscope at an accelerating voltage of 200 kV. ICP results were measured by a PerkinElmer Optima 4300DV spectrometer. TG/DTG was performed using a Netzsch STA 449F3 simultaneous thermal analyzer at a heating rate of 2 °C min$^{-1}$ in Ar. FT-IR transmittance spectra were measured using a 60-SXB IR spectrometer with paraffin oil as the dispersant. XPS was recorded with a VG Multilab 2000.

**Electrochemical measurements.** To conduct the electrochemical measurements, 2016 coin cells were assembled in a glove box filled with pure Ar gas. The working electrodes were prepared by mixing 70% active material, 20% acetylene black and 10% carboxyl methyl cellulose binder, and spreading mixture on a copper foil. The electrodes were cut into small wafers with a diameter of 1.0 cm. The mass loading of active material was 1.0–2.0 mg cm$^{-2}$. For the sodium half cells, Na discs were used as both the counter and reference electrodes. The electrolyte was composed of 1 M $NaClO_4$ dissolved in a mixture of ethylene carbonate (EC)/dimethyl carbonate (DMC) (1:1 w/w) with 5% fluoroethylene carbonate (FEC), and a Whatman glass microfiber filter (Grade GF/F) was used as the separator. The Na-ion full cells were fabricated using the same separator and electrolyte as the half cells, with CVO-450 as the anode and $Na_3V_2(PO_4)_3$/C nanoparticles as the cathode. Galvanostatic charge/discharge measurements were performed with a multi-channel battery testing system (LAND CT2001A). CV and EIS were recorded with an electrochemical workstation (Autolab PGSTAT 302 N). For the in situ XRD test, the cell was assembled with an X-ray-transparent beryllium window as the current collector, and the electrode was placed behind it. The cell was cycled at a current density of 200 mA g$^{-1}$, meanwhile, the in situ XRD patterns were collected with a planar detector every 3 min.

**Ab initio calculations.** We have used the first-principles pseudopotential plane-wave method based on the DFT incorporated into the CASTEP computational code[63]. The exchange and correlation potentials are described in the framework of the generalized gradient approximation[64]. A plane-wave basis was employed with an energy cutoff of 500 eV. All geometries were optimized and total energy calculated, convergence criteria were set to $5 \times 10^{-6}$ eV and 0.01 eV·Å$^{-1}$ for energy and force, respectively.

**Data availability.** The authors declare that the data supporting the findings of this study are available within the paper and its Supplementary Information files and can be requested by writing to the corresponding authors.

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

## Acknowledgements

This work was supported by the National Key Research and Development Program of China (2016YFA0202603), the National Basic Research Program of China (2013CB934103), the Programme of Introducing Talents of Discipline to Universities (B17034), the National Natural Science Foundation of China (51521001, 11404230), the National Natural Science Fund for Distinguished Young Scholars (51425204), Foundation of Science and Technology Bureau of Sichuan Province (2013JY0085) and the Fundamental Research Funds for the Central Universities (WUT: 2016III001, 2017III009). This work is supported by the project of Innovative group for low cost and long cycle life Na ion batteries R&D and industrialization of Guangdong Province (Grant No. 2014ZT05N013). L.M. gratefully acknowledged the financial support from China Scholarship Council (No. 201606955096). Special thanks are extended to Professor Xuedong Bai of the Institute of Physics, Chinese Academy of Science for his strong support of the in situ TEM measurement and Professor Dongyuan Zhao of Fudan University for his strong support and stimulating discussions.

## Author contributions

X.X. and L.M. proposed and designed the research. X.X., C.N., X.W., L.P., L.H., and W.R. performed the materials synthesis and electrochemical analyses. X.X., C.S., and J.M. conducted the materials characterization. X.W. and J.W. performed the in situ XRD characterization and assembled the single nanowire devices. B.S. and M.D. designed and conducted the ab initio calculations. X.X., L.M., and C.N. co-wrote the paper. All authors discussed the results and commented on the manuscript.

## Additional information

**Competing interests:** The authors declare no competing financial interests.

