## [Peer Review File · Nature Communications]

Reviewers' comments:

Reviewer #1 (Remarks to the Author):

This manuscript is about synthesis of CaV₄O₉ (Alkaline earth metal family) nanowires and performance as working electrode in sodium-ion battery half cell. The nanowires were synthesized by use of a hydrothermal process and their morphology and chemical structure was established by using a range of spectroscopic and microscopy techniques. When tested as working electrode the nanowires show sodium charge capacity of approx. 300 mAh/g. Cycling performance for 1600 cycles is reported. Further, some ab-initio calculations, ex-situ XRD, XPS, and TEM data at various stages of Na-insertion is reported. Use as working electrode and detailed microstructural characterization is perhaps the main contribution of this work.

Although the reviewer found the chemical/morphological characterization of synthesized nanowire to be comprehensive, the claims related to electrochemical performance are somewhat unfounded or exaggerated and require further investigation. The manuscript in its present form may not be suitable for nature communications.

1. Sodium ion batteries (compared to LIB) are probably suitable for large scale energy storage (MWh, GWh) applications only which would required kilograms or tons of electrode material. Nanowire morphology is unsuitable for such applications due to scalability and cost issues. Authors are directed to recent review article by Tarascon:

<http://www.nature.com/nchem/journal/v7/n1/abs/nchem.2085.html>

2. Nanowire morphology generally shows high capacity (sometimes even independent of the chemical structure) due to increased surface effects etc. Such materials also show poor volumetric capacity. Authors should include areal capacity and compare the performance of their material with those (low cost) from literature (For example, <http://www.nature.com/articles/ncomms5033> and <http://pubs.acs.org/doi/abs/10.1021/jp5080847>).

3. Equation 1 and 2: claim has been made that the mechanism is different than conversion reaction. How? the claim is not convincing. Metallic Calcium may have oxidized to CaO.

4. Figure 3d: capacity is not stable. Coulombic efficiency not included in the graph. The material is therefore not promising as battery electrode. The current density is too high which means the test was performed in less time. Long term testing i.e., days/months is needed.

5. Figure 3a: the materials shows large voltage hysteresis (energy inefficiency) which would lower the operating voltage in full cell. This effect need to be discussed in detail.

Reviewer #2 (Remarks to the Author):

The manuscript presented the alkaline earth metal CaV₄O₉ nanowires were fabricated and shown the exciting electronic conductivity, applicable capacity, good rate capability and outstanding long-term cycling stability. Moreover, the specific Na⁺ storage mechanism was proposed and demonstrated through in situ/ex situ characterization and theoretical calculation. The results were impressive and inspiring. However, the manuscript may need some improvement before it is accepted.

1. There are inconformity of terminology happened in the manuscript such as "charge-discharge", which is in the 52th line while "charge/discharge" present in the 135th line; besides, add the page numbers in the manuscript may be more better ; etc.
2. In the manuscript, the author mentioned " The XRD results show that the sample prepared at 450 °C (marked as CVO-450) displays much weaker diffraction peaks than the sample prepared at 550 °C (marked as CVO-550), indicating the low crystallinity of CVO-450". Moreover, CVO-450 and CVO-550 display the larger difference of electrochemical performance, so the phase transformation is crucial to the whole process, we suggest XRD patterns of different sintering temperature should be displayed to investigate the whole process.
3. In the 128th line, the author ascribe the irreversible cathodic peak at ~1.1 V of CVO-550 in the initial scan to "irreversible structural dissociation in the initial sodiation process due to its high crystallinity", why high crystallinity can lead irreversible structural dissociation, this may be not very reasonable or author should explain it clearly.
4. The rate performances of CVO-450 is better than CVO-550 while CVO-550 show more better long cycle performance than CVO-450, we suggest the author should further explain the possible reasons.

Reviewer #3 (Remarks to the Author):

Recommendation: Accept with minor revisions.

The submitted manuscript is clearly written and well-organized with good presentation of the results obtained and complete discussion and, in my opinion, meets the general criteria for publishing in Nature Communications. It is plausible that this work includes computational modeling in addition to the principal experimental part. The conclusions are validated by a number of experiments that used different methods and the DFT predictions. The authors have performed a very good research work and I have only a few minor comments:

1. The authors refer to the larger ionic size of Na to explain the low performance of SIBs (page 2). However, Liu et al. (PNAS 2016, 113, 3735–3739) showed that the low capacity of SIBs is rather related to the competition between trends in the ionization energy and the ion-substrate bonding.
2. It is unclear for me why the formation mechanism that works for CVO-450: "the evaporation of the crystal water leads to the formation of low-crystalline (amorphous) structure and the cavities on the nanowires (Supplementary Fig. 6)" (pages 4-5), does not work for CVO-550 and this material is characterized by the high crystallinity? It would be nice to have a discussion about this in the paper.
3. Actually, this comment is related to comment 2. To explore the formation mechanism of the low-crystalline structure of the CVO-450 nanowires, TG/DTG, XRD and FT-IR measurements were performed (Supplementary Figs. 4, 5). However, the authors do not show and discuss TG/DTG for CVO-550. This should definitely be done and compared to that for CVO-450.

Reviewer #4 (Remarks to the Author):

The paper entitled "Alkaline earth metal vanadates as new sodium-ion battery anodes" reports the electrochemical performance and reaction mechanism of CaV₄O₉ nanowires for sodium ion batteries. It is remarkable that CaV₄O₉ nanowires exhibited stable capacity retention over 1600 cycles. However, this paper is not appropriate for the publication in Nature Communications because of lack of novelty.

1. CaV4O9 is irreversibly decomposed into CaO and VO2 through a conversion reaction in the first cycle, and Na⁺ ions are inserted into VO2 in the subsequent cycles. This means that VO2 is a real active material, although CaV4O9 is a starting material. However, VO2 is well known as anode materials for sodium ion batteries, and many groups studied their electrochemical performance and reaction mechanism.[1,2] In other words, CaV4O9 is not a practically new and novel material and can be considered as a mixture of CaO and VO2, although CaV4O9 is first examined for sodium ion batteries in this paper. In addition, CuV2O6 showing a similar conversion reaction mechanism has already introduced as an anode for lithium ion batteries.[3,4]

[1] Steffen Hart et al., Vanadium-based polyoxometalate as new material for sodium-ion battery anodes, *Journal of Power Sources*, 288, (2015) 270–277

[2] C. Didier et al., Electrochemical Na-Deintercalation from NaVO2, *Electrochem. Solid-State Lett.* 14, (2011) A75-A78

[3] P. Prahasini et al., A novel attempt for employing brannerite type copper vanadate as an anode for lithium rechargeable batteries, *J Mater Sci: Mater Electron* 27, (2016) 3292–3297

[4] Kangning Zhao, Graphene Oxide Wrapped Amorphous Copper Vanadium Oxide with Enhanced Capacitive Behavior for High-Rate and Long-Life Lithium-Ion Battery Anodes, *Adv. Sci.* 2, (2015) 1500154

2. The specific capacity and average redox potential of CaV4O9 are about 300 mAh/g and about 1 V vs. Na/Na⁺. Considering the energy density of sodium ion batteries, CaV4O9 is not an attractive anode material because of its poor specific capacity and high average redox potential.

3. The authors insisted that the volume change of CaV4O9 is small (~10%) as compared to P, Sn, and Sb. However, this comparison is not fair, because the specific capacities of P, Sn, and Sb are much larger than that of CaV4O9. The volume change of CaV4O9 should be compared with that of VO2 or hard carbon delivering similar reversible capacity (about 300 mAh/g). For hard carbon, there is no volume change during charge and discharge.

4. The authors explained that the improved cycle performance of CaV4O9 is attributed to the buffering role of CaO. However, the volume change of VO2 itself is not large during charge and discharge. Therefore, it is difficult to find what the buffering role of CaO is. Of course, it is okay to say that the buffering role of CaO contributes to capacity retention of metal oxides such as CaO+SnO showing large volume change, as the authors cited. Therefore, the buffering role of CaO should be intensively studied, because this phenomenon is one of the most important aspects of CaV4O9.

5. CaV4O9 delivered higher specific capacity than its theoretical value, and the authors explained that the extra capacity is attributed to the Na⁺ storage in the amorphous region of CaV4O9 or the residual water or OH groups on the surface of the nanowires. However, these claims were not supported by any evidence, although the extra capacity is an important abnormal behavior of CaV4O9. Also, the authors used large amount of carbon additive (about 30 wt.% of active material) for the electrode preparation, and carbon additive is known to deliver the reversible capacity of about 250 mAh/g. This means that about 70 mAh/g of specific capacity can be contributed from carbon additive. This amount of specific capacity is similar to that of the extra capacity of CaV4O9 in this report.

6. The authors insisted that the good electrical conductivity of CaV4O9 contributes to improve electrochemical performance. However, CaV4O9 is irreversibly decomposed into CaO and VO2 through conversion reaction in the first cycle. Therefore, little CaV4O9 exists in the subsequent cycles, indicating that CaV4O9 can not contribute the improved electrochemical performance.

Reviewers' comments:

Reviewer #1 (Remarks to the Author):

The revised manuscript is improvement over previous version. However, wild claims need to be removed before it can be accepted for publication:

1. Abstract: Remove the statement: "which result in its long-term cycling stability over 1,600 cycles." The test for 1600 cycles was performed at high current which means short time. There is no evidence that the material is stable over long term.
2. There is no reason to believe that the material is scalable and hence claims regarding large scale energy storage need to be removed: Abstract: "large-scale energy storage".
3. Figure 3b. Conductivity of the powder is more important than individual nanowire. Four point conductivity of the nanowire powder (or pressed pellet) is recommended.

Reviewer #2 (Remarks to the Author):

The authors have issued all the concerns, so I think it is suitable to publish the paper in NC.

Reviewer #3 (Remarks to the Author):

I am satisfied with the response provided by the authors to my comments and recommend this manuscript for publication in "Nature Communications".

Reviewer #4 (Remarks to the Author):

The authors have addressed all of my previous concerns, and their revisions have substantially improved the manuscript. I agree that the proposed conversion reaction mechanism of CaV4O9 is new and interesting. However, some issues are not still clear.

1. CaV4O9 showed low reversible capacity (less than 350 mAh/g), poor coulombic efficiency for the initial cycle (55~70%), and high redox potential (0~3 V). These properties result in poor energy density of full cells. Based on that Li4Ti5O12 with a high redox potential (~1.5 V vs. Li/Li+) is commercialized, the authors insisted that the high redox potential of CaV4O9 is not a problem for commercialization. However, CaV4O9 shows slopping voltage profiles, while Li4Ti5O12 has a plateau profile at ~1.5 V vs. Li/Li+. In the other words, the reversible capacity of CaV4O9 delivered in the voltage range of 1.5~3.0 V is useless, because the working voltage of full cells is too low to operate a device when the redox potential of anode is between 1.5~3.0 V. This indicates that the practical reversible capacity of CaV4O9 as anode is only about 200 mAh/g. In addition, the poor coulombic efficiency of the initial cycle (55~70%) will further decrease the energy density of full cells. Therefore, CaV4O9 is not practically promising, although the cycle performance of CaV4O9 is excellent.
2. The role of CaO is not still clear. Alloy materials with high volume changes (>100 %) during charging and discharging are known to show poor cycle performance because of pulverization and

repetitive electrolyte decomposition during cycling. However, NaVO₂ shows small volume changes (~10%) during charging and discharging, suggesting that the pulverization will not occur during cycling. This means that NaVO₂ does not require the buffering or preserving role of CaO.

REVIEWERS' COMMENTS:

Reviewer #1 (Remarks to the Author):

Comments have been addressed to satisfaction.

Reviewer #4 (Remarks to the Author):

The authors have addressed essentially all of my previous concerns, and it is appropriate for the publication in Nature Communications.

Respond to Reviewer #1:

This manuscript is about synthesis of CaV_4O_9 (Alkaline earth metal family) nanowires and performance as working electrode in sodium-ion battery half cell. The nanowires were synthesized by use of a hydrothermal process and their morphology and chemical structure was established by using a range of spectroscopic and microscopy techniques. When tested as working electrode the nanowires show sodium charge capacity of approx. 300 mAh/g. Cycling performance for 1600 cycles is reported. Further, some ab-initio calculations, ex-situ XRD, XPS, and TEM data at various stages of Na-insertion is reported. Use as working electrode and detailed microstructural characterization is perhaps the main contribution of this work.

Although the reviewer found the chemical/morphological characterization of synthesized nanowire to be comprehensive, the claims related to electrochemical performance are somewhat unfounded or exaggerated and require further investigation. The manuscript in its present form may not be suitable for nature communications.

Our response:

We thank the reviewer for the thoughtful review and critical comments about our manuscript. We welcome the opportunity to address and clarify the issues raised in the reviewer's report, and believe that the additional experiments and revisions substantially address the reviewer's comments and strengthen our revised manuscript. Our responses to the points raised in the report are as follows:

1. Sodium ion batteries (compared to LIB) are probably suitable for large scale energy storage (MWh, GWh) applications only which would required kilograms or tons of electrode material. Nanowire morphology is unsuitable for such applications due to scalability and cost issues. Authors are directed to recent review article by Tarascon: <http://www.nature.com/nchem/journal/v7/n1/abs/nchem.2085.html>

Response to comment 1:

We thank the reviewer for the comments together with the provided reference. We have cited this important reference in the revised manuscript (Ref. 3, revised submission). And before giving response to the reviewer's comments, we would like

to state our viewpoint that the realization of a new technology in practical application cannot be accomplished in one action, including sodium ion battery technology. As researchers, we always start with the fundamental research.

This work is definitely focused on the fundamental study of CaV_4O_9 . As we pointed out in the original manuscript, our motivation to fabricate nanowires is because of the unique advantages of nanowires in energy storage and *in situ* characterization. As we can see, based on the advantages of nanowires, we assembled a single nanowire device and performed *in situ* TEM measurement, then manifested the intrinsic high conductivity and small volume change of this material, which together with the superior electrochemical performance and specific reaction mechanism, indicate that CaV_4O_9 could be a promising electrode material for sodium ion batteries. Considering that CaV_4O_9 has never been investigated for sodium ion battery in previous reports, we believe these results are significant and will lay a strong foundation for the further investigation on CaV_4O_9 and other alkaline earth metal vanadates, and then will make positive impacts on the development of sodium ion batteries.

In the future investigations, we can develop other methods to synthesize CaV_4O_9 with suitable morphology and high yields. Actually, we are carrying out the relevant research following the fundamental results of this work.

2a. Nanowire morphology generally shows high capacity (sometimes even independent of the chemical structure) due to increased surface effects etc.

Response to comment 2a:

Thank you for the comment. Based on the reviewer's comment, we tested the specific surface area of the CVO-450 and CVO-550 nanowires using nitrogen adsorption-desorption techniques. The data are shown below. The BET specific surface areas of CVO-450 and CVO-550 nanowires were measured to be $21.1 \text{ m}^2 \text{ g}^{-1}$ and $7.7 \text{ m}^2 \text{ g}^{-1}$, respectively. The low surface areas of the obtained nanowires indicate the negligible surface effects to the capacity.

Nitrogen adsorption-desorption isotherms and pore size distribution (inset) of CVO-450 (a) and CVO-550 (b).

2b. Such materials also show poor volumetric capacity. Authors should include areal capacity and compare the performance of their material with those (low cost) from literature (For example, <http://www.nature.com/articles/ncomms5033> and <http://pubs.acs.org/doi/abs/10.1021/jp5080847>).

Response to comment 2b:

We thank the reviewer very much for the comments and the provided references. Areal capacity can be calculated based on the mass capacity and mass loading as follows:

$$C_{\text{areal}} = \frac{A \times M \times C_{\text{mass}} \times 10^{-3}}{A} = C_{\text{mass}} \times M \times 10^{-3}$$

Where C_{areal} (mAh cm⁻¹) represents areal capacity, C_{mass} (mAh g⁻¹) represents mass capacity, A (cm²) represents area of the electrode, M (mg cm⁻²) represents the mass loading.

The areal capacities of CVO-450 and CVO-550 were calculated based on the galvanostatic charge/discharge data at 100 mA g⁻¹ (Supplementary Figure 7e in our original submission). The comparison of the areal capacity between this work and other carbon-based anodes are shown as follows:

Materials	C_{mass} (mAh g ⁻¹)	M (mg cm ⁻²)	Current density (mA g ⁻¹)	C_{areal} (mAh cm ⁻¹)	Reference
CVO-450	~350	~1.0	100	~0.35	This work
CVO-550	~290	~1.1	100	~0.32	This work
Expanded graphite	~300	~0.5	20	~0.15	R1 (Provided by the reviewer)
rGO paper	~200	~0.75	20	~0.15	R2 (Provided by the reviewer)
Hard carbon	~250	~1.5	12	~0.37	R3
Carbon nanofibers	~255	~1.0	40	~0.25	R4

Our preliminary results show that even though at a higher current density, the areal capacities of CVO-450 and CVO-550 in this work are comparable to or higher than that of some carbon-based materials, indicating the superiority of CaV₄O₉ in areal capacity and rate capability.

By increasing the mass loading of active material, the areal capacity of CaV₄O₉ can be further improved. Besides, the areal capacity is also relevant to the morphology and particle size of the material. As pointed out in our response to comment 1, we are carrying out new experiments to synthesize CaV₄O₉ with suitable morphology and high yields. We believe that the areal capacity of CaV₄O₉ can be improved significantly in our next investigation.

3. Equation 1 and 2: claim has been made that the mechanism is different than conversion reaction. How? the claim is not convincing. Metallic Calcium may have oxidized to CaO.

Response to comment 3:

We thank the reviewer for the comments. Based on the previous references^{R5-R8}, the typical conversion reactions, which are always observed in Fe/Mn/Co/Cu-based

anodes, are as follows:

where M represents transition metal, X represents anion, and n represents formal oxidation state of X.

The characteristics of the typical conversion reaction are that pure metal (such as Fe) and Na_2O will be generated after initial Na^+ insertion. But for CaV_4O_9 , as indicated by equation 1 and 2, neither metallic V nor Na_2O are formed during the Na^+ insertion/extraction process. Meanwhile, as demonstrated by the XPS results (Supplementary Figure 14, original submission), also no metallic Ca was generated both in the sodiation and desodiation state. Ca^{2+} or CaO is electrochemically inactive due to the strong Ca-O bond strength and cannot be reduced to metallic Ca.

Therefore, we agree that the Na^+ insertion into CaV_4O_9 results in a conversion reaction, but this conversion is different from the typical conversion reaction observed in other transition metal oxides. To further clarify this point, the corresponding statement in page 10 has been revised as follows:

There is no metallic V or Na_2O generated both in the initial discharge process and the subsequent cycles, which is different from the typical conversion reaction observed in other transition metal oxides.

4a. Figure 3d: capacity is not stable. Coulombic efficiency not included in the graph. The material is therefore not promising as battery electrode.

Response to comment 4a:

We thank the reviewer for the comments. The capacity fluctuation is a frequent phenomenon in sodium-ion half cells, which has also been observed in many previous reports^{R5,R8-R10}. This phenomenon may be related to the high activity of the metallic Na. Based on the reviewer's comments, we have retested the cycling performance of CVO-450 and VO₂-450 at current density of 1000 mA g⁻¹, and replaced the fluctuant data in the original manuscript. We have updated Figure 3d and the corresponding data in the revised manuscript, and the Coulombic efficiency has also been added. The new Figure 3d is shown as follows:

New Figure 3d

We believe the capacity fluctuation is not an intrinsic property of the electrode material, and it will be effectively alleviated in the case of sodium-ion full battery.

As to the potential of a material as battery electrode, a comprehensive consideration of capacity, rate capability, cycle life, cost and safety is needed to judge whether it is promising or not. For CaV_4O_9 in this manuscript, our preliminary results indicate that it has an applicable capacity, good rate capability, excellent cycling stability and moderate voltage. Considering the good electrochemical properties together with the low cost and abundant resources of Ca and V, we believe that CaV_4O_9 or other alkaline earth metal vanadates have a bright prospect and are promising for sodium-ion battery or other battery electrodes.

4b. The current density is too high which means the test was performed is less time. Long term testing i.e., days/months is needed.

Response to comment 4b:

Thank you for the comment. The electrochemical performance at a low current density (100 mA g^{-1}) was also displayed in Supplementary Figure 7 of the original manuscript (as shown below). The testing for 100 cycles at 100 mA g^{-1} lasted for over one month. Besides, CV measurements were also performed at a low rate of 0.1 mV s^{-1} for 3 cycles, and the testing lasted for over 3 days.

Supplementary Figure 7 | Electrochemical performance of CVO-450 and CVO-550. (a,b) CV curves of CVO-450 and CVO-550 at the scan rate of 0.1 mV s^{-1} in the voltage range from 0.01 to 3.0 V *versus* Na^+/Na . (c,d) Discharge-charge profiles of CVO-450 and CVO-550 after different cycles at 100 mA g^{-1} . (e) Cycling performance of CVO-450 and CVO-550 at current density of 100 mA g^{-1} . (f) Comparison of the coulombic efficiency between CVO-450 and CVO-550 at

current density of 100 mA g^{-1} .

In addition, we also added the electrochemical test of CVO-450 at a very low current density (10 mA g^{-1}), in which case, every discharge or charge test lasts for ~ 40 h. The discharge/charge profiles are shown below, and they do not show any voltage plateau in the whole processes, which is consistent with our previous results.

5. Figure 3a: the materials shows large voltage hysteresis (energy inefficiency) which would lower the operating voltage in full cell. This effect need to be discussed in detail.

Response to comment 5:

We thank the reviewer for the comments and suggestions. Voltage hysteresis is a general challenge for conversion-based anodes, which has also been observed in most previous reports about SIB anodes^{R5-R8}.

We have carefully taken the reviewer's suggestion into consideration, and did some necessary revisions in the manuscript.

In page 5, we added discussions about the voltage hysteresis and a relative reference in regarding to Figure 3a, the discussions is as follows:

The voltage hysteresis between discharge and charge, which may be attributed to the different reaction paths during sodiation and desodiation⁵⁷, shows a minor change from the 2nd cycle to the 100th cycle, indicating a good electrochemical stability.

In page 7, the discussions about the full battery have also been revised as follows:

Full batteries with higher output voltage and smaller voltage hysteresis can be produced by using cathodes with higher voltages and optimizing the transport properties of both electrodes.

Reference:

- R1. Wen, Y. *et al.* Expanded graphite as superior anode for sodium-ion batteries. *Nat. Commun.* **5**, 4033 (2014).
- R2. David, L. & Singh, G. Reduced graphene oxide paper electrode: opposing effect of thermal annealing on Li and Na cyclability. *J. Phys. Chem. C.* **118**, 28401 (2014).
- R3. Bai, Y. *et al.* Hard Carbon Originated from Polyvinyl Chloride Nanofibers As HighPerformance

- Anode Material for Na-Ion Battery. *ACS Appl. Mater. Interfaces* **7**, 5598 (2015).
- R4. Luo, W. *et al.* Carbon nanofibers derived from cellulose nanofibers as a long-life anode material for rechargeable sodium-ion batteries. *J. Mater. Chem. A* **1**, 10662 (2013).
- R5. Zhang, N. *et al.* 3D porous γ -Fe₂O₃@C nanocomposite as high-performance anode material of Na-Ion batteries. *Adv. Energy Mater.* **5**, 1401123 (2014).
- R6. Liu, Y., Zhang, N., Yu, C., Jiao, L. & Chen, J. MnFe₂O₄@C nanofibers as high-performance anode for sodium-ion batteries. *Nano Lett.* **16**, 3321–3328 (2016).
- R7. Rahman, M. M. *et al.* Ex situ electrochemical sodiation/desodiation observation of Co₃O₄ anchored carbon nanotubes: a high performance sodium-ion battery anode produced by pulsed plasma in a liquid. *Nanoscale* **7**, 13088-13095 (2015).
- R8. Yuan, S. *et al.* Engraving Copper Foil to Give Large-Scale Binder-Free Porous CuO Arrays for a High-Performance Sodium-Ion Battery Anode. *Adv. Mater.* **26**, 2273–2279 (2014).
- R9. Li, W., *et al.* A high performance sulfur-doped disordered carbon anode for sodium ion batteries. *Energy Environ. Sci.* **8**, 2916-2921 (2015).
- R10. Liu, J., *et al.* Uniform yolk-shell Sn₄P₃@C nanospheres as high-capacity and cycle-stable anode materials for sodium-ion batteries. *Energy Environ. Sci.* **8**, 3531-3538 (2015).

Respond to Reviewer #2:

The manuscript presented the alkaline earth metal CaV_4O_9 nanowires were fabricated and shown the exciting electronic conductivity, applicable capacity, good rate capability and outstanding long-term cycling stability. Moreover, the specific Na^+ storage mechanism was proposed and demonstrated through in situ/ex situ characterization and theoretical calculation. The results were impressive and inspiring. However, the manuscript may need some improvement before it is accepted.

Our response:

We thank the reviewer very much for his/her high evaluation and strong support on our work. We also thank you for the time and efforts throughout the process. We have made further revisions in the manuscript based on the reviewer's comments and suggestions. Our point-by-point responses are listed below:

1. There are inconformity of terminology happened in the manuscript such as "charge-discharge", which is in the 52th line while "charge/discharge" present in the 135th line; besides, add the page numbers in the manuscript may be more better ; etc.

Response to comment 1:

Thank you very much for the kind reminding and suggestion. We have unified the terminology as "charge/discharge" throughout the revised manuscript, and we have also added the page numbers in the revised manuscript.

2. In the manuscript, the author mentioned " The XRD results show that the sample prepared at 450 °C (marked as CVO-450) displays much weaker diffraction peaks than the sample prepared at 550 °C (marked as CVO-550), indicating the low crystallinity of CVO-450". Moreover, CVO-450 and CVO-550 display the larger difference of electrochemical performance, so the phase transformation is crucial to the whole process, we suggest XRD patterns of different sintering temperature should be displayed to investigate the whole process.

Response to comment 2:

We thank the reviewer for the comments. Actually, the XRD patterns of different sintering temperature have been displayed in the Supplementary Figure 5a (as shown below) in our original submission. We also made corresponding discussions below Supplementary Figure 5. To cite the text from original Supplementary Information:

XRD and FT-IR measurements were further performed on the samples after different sintering conditions (Supplementary Fig. 5). XRD results show that the strong peak at about 11° for the unsintered samples vanished gradually as the sintering temperature increase. Notably, the sample prepared at 400 °C is basically an amorphous state. When the temperature increases to 450 °C for 8h, the sample recrystallizes and forms low crystalline CaV_4O_9 (CVO-450). From FT-IR spectra, the stretching vibration and bending vibration of H_2O were clearly observed for the sample before annealing, but disappeared for the annealed sample at 450 °C for 8 h, consistent well with the XRD results. Based on the TG/DTG, XRD and FT-IR results, the formation mechanism of the low crystalline structure together with the cavities of CVO-450 nanowires was proposed as illustrated in Supplementary Fig. 6. The unsintered sample is a layered structure with the Ca^{2+} and H_2O molecules

distributed in the interlayers. As the sintering temperature increases, the H₂O molecules gradually evaporate, which leads to the formation of cavities. Meanwhile, the loss of H₂O molecules in the interlayers results in the collapse of the layered structure and the decrease of the crystallinity. When the sintering condition was set at 450 °C for 8h, almost all the H₂O molecules evaporate and the structure recrystallizes to form low crystalline CaV₄O₉ nanowires (CVO-450). When the temperature increases to 550 °C, the higher temperature push the diffusion of the atoms (mass transfer process) to form a stable crystal phase, leading to the higher crystallinity and the closure of the cavities together with the irregular morphology of CVO-550.

Supplementary Figure 5 | (a) XRD patterns of the Ca-V-O nanowire samples at different annealing condition. (b–d) FT-IR spectra of Ca-V-O nanowire samples before annealing (b), sintered at 400 °C for 5 h (c) and sintered at 450 °C for 8 h (d), respectively. (For the FT-IR spectra, the paraffin oil was used as the dispersant rather than KBr to avoid the moisture absorption during the test.)

To further clarify this process, we have also revised Supplementary Fig. 6 as below:

Supplementary Figure 6 | Illustration of the formation mechanism of the CVO-450 and CVO-550 nanowires.

3. In the 128th line, the author ascribe the irreversible cathodic peak at ~1.1 V of CVO-550 in the initial scan to “irreversible structural dissociation in the initial sodiation process due to its high crystallinity”, why high crystallinity can lead irreversible structural dissociation, this may be not very reasonable or author should explain it clearly.

Response to comment 3:

Thank you very much for the kind comment. As demonstrated by *in situ* XRD and *ex situ* TEM results in the manuscript, the Na⁺ insertion leads to the amorphization and dissociation of the CaV₄O₉ structure. For CVO-550, it shows much higher crystallinity than CVO-450, therefore, the structural dissociation of CVO-550 will result in more distinct irreversible cathodic peak.

To avoid misunderstanding, we have revised the corresponding sentence in the revised manuscript, showing as follows:

CVO-450 shows no obvious redox peaks, but CVO-550 displays an irreversible cathodic peak at ~1.1 V in the initial scan (Supplementary Fig. 7a,b), which may correspond to the irreversible structural dissociation in the initial sodiation process.

4. The rate performances of CVO-450 is better than CVO-550 while CVO-550 show more better long cycle performance than CVO-450, we suggest the author should further explain the possible reasons.

Response to comment 4:

We thank the reviewer very much for the kind comments.

The better rate performance of CVO-450 is attributed to its low crystallinity. Previous studies (Refs. 42, 46, revised manuscript) have shown that the amorphous or less-crystalline structure provides more active sites and better ion diffusion kinetics compared with the high crystalline structure. Therefore, the low-crystalline CVO-450 shows better rate performance than CVO-550. We have discussed this point in our original submission. To cite the text from the original manuscript in page 8:

Moreover, the amorphous or less-crystalline structure provides more active sites and better ion diffusion kinetics^{42,46}, which explains the significant increase in capacity for CVO-550 in the initial cycles. And due to the low crystallinity of CVO-450 in the pristine state, it displays better rate capability than CVO-550.

For the long cycling performance, it is proposed that the inferior cycling stability of CVO-450 is because of the residual water on the nanowires. As shown in Supplementary Fig. 4-6, before annealing, the nanowires contain considerable crystal water (~6.2%). Based on the XRD results and FT-IR spectra in Supplementary Fig. 5, the crystal water did not fully evaporate after being annealed at 400 °C (CVO-400). Thus, it is reasonable that minor residual water will also exist in the sample annealed at 450 °C (CVO-450). As reported by Grey’s group (Ref. 56, revised manuscript), residual water or OH groups can act as a major source of additional capacity. However, residual water also leads to the consumption and decomposition of the electrolyte and does not benefit the long-term cycle life of the batteries, which explains why CVO-450 has inferior cycling stability compared to CVO-550.

To further give evidence to this point, we tested the performance of CVO-400, and make comparison with CVO-450 and CVO-550, as below. The higher capacity at initial stage but much poorer cycling stability of CVO-400 further confirms our analyses above.

Supplementary Figure 17 | Comparison of specific capacity and cycling stability of CVO-400, CVO-450 and CVO-550 at current density of 1000 mA g⁻¹.

Based on the reviewer's suggestions, we revised the corresponding discussions in page 12 to further clarify this point. The revised discussions are as follows:

The extra capacity can be attributed to the residual water in the nanowires. As described before, the nanowire samples contain considerable crystal water (~6.2%) before annealing (Supplementary Fig. 4-6). As indicated by the XRD and FT-IR results (Supplementary Fig. 5a and 5c), the crystal water was still not fully removed for CVO-400 (annealed at 400 °C); thus, it is reasonable that there is minor residual water in CVO-450. As reported by Grey's group⁵⁶, residual water or OH groups can act as a major source of additional capacity. However, residual water also leads to the consumption and decomposition of the electrolyte and does not benefit the long-term cycle life of the batteries, which explains why CVO-450 has inferior cycling stability compared to CVO-550 (Fig. 3d). To further confirm this point, CVO-400 was also tested as electrode and compared with CVO-450 and CVO-550. As expected, the CVO-400 exhibits higher initial capacity but much poorer cycling stability (Supplementary Fig. 17).

Respond to Reviewer #3:

Recommendation: Accept with minor revisions.

The submitted manuscript is clearly written and well-organized with good presentation of the results obtained and complete discussion and, in my opinion, meets the general criteria for publishing in Nature Communications. It is plausible that this work includes computational modeling in addition to the principal experimental part. The conclusions are validated by a number of experiments that used different methods and the DFT predictions. The authors have performed a very good research work and I have only a few minor comments:

Our response:

We thank the reviewer very much for his/her high evaluation and kind comments on our manuscript. We have taken your comments and suggestions meticulously and made further revisions on the manuscript. Our point-by-point responses are listed below:

1. The authors refer to the larger ionic size of Na to explain the low performance of SIBs (page 2). However, Liu et al. (PNAS 2016, 113, 3735–3739) showed that the low capacity of SIBs is rather related to the competition between trends in the ionization energy and the ion-substrate bonding.

Response to comment 1:

Thank you very much for your kind comment and the important reference. It is true that several different reasons have been proposed to explain the inferior performance of SIBs compared to LIBs. And the larger ionic size of Na⁺ is one of the most widely accepted reasons so far. Therefore, we state that “The problem may be due to the larger ionic size of Na⁺, which results in sluggish reaction kinetics or severe degradation of the electrodes.”

Considering the high relevance of this reference provided by the reviewer, we have cited this paper (Ref. 10, revised manuscript) and revised the sentence as follows:

The problem may be due to the larger ionic size of Na⁺ or generally weak binding of Na with substrate, which results in sluggish reaction kinetics or severe degradation of the electrodes⁸⁻¹⁰.

2. It is unclear for me why the formation mechanism that works for CVO-450: “the evaporation of the crystal water leads to the formation of low-crystalline (amorphous) structure and the cavities on the nanowires (Supplementary Fig. 6)” (pages 4-5), does not work for CVO-550 and this material is characterized by the high crystallinity? It would be nice to have a discussion about this in the paper.

Response to comment 2:

We thank the reviewer very much for the kind comment. For CVO-550, the sample was annealed at 550 °C, in which condition, the crystal water will be fully evaporated quickly (as indicated by the TG curve in Supplementary Fig. 4). And the high temperature (550 °C) will also push the diffusion of the atoms (mass transfer process)

and increase the crystallinity of the sample to form a stable crystal phase. Thus, CVO-550 displays higher crystallinity than CVO-450. Meanwhile, the mass transfer process will lead to the closure of the cavities on the nanowires together with the irregular morphology (as indicated by the TEM images in Figure 2i and 2j). To better illustrate the whole processes, we have revised Supplementary Fig. 6 as follows:

Supplementary Figure 6 | Illustration of the formation mechanism of the CVO-450 and CVO-550 nanowires.

The corresponding discussions in Supplementary information about Supplementary Figure 6 have been revised as follows:

When the temperature increases to 550 °C, the higher temperature push the diffusion of the atoms (mass transfer process) to form a stable crystal phase, leading to the higher crystallinity and the closure of the cavities together with the irregular morphology of CVO-550.

We also added some discussion about this in page 5 of the manuscript, and the revised discussion is as follows:

These results demonstrated that the evaporation of crystal water leads to the formation of the low-crystalline structure and the cavities on the CVO-450 nanowires (Supplementary Fig. 6). While for CVO-550, the higher annealing temperature leads to the higher crystallinity and the closure of the cavities together with the irregular morphology. Details can be found in the Supplementary Information.

3. Actually, this comment is related to comment 2. To explore the formation mechanism of the low-crystalline structure of the CVO-450 nanowires, TG/DTG, XRD and FT-IR measurements were performed (Supplementary Figs. 4, 5). However, the authors do not show and discuss TG/DTG for SVO-550. This should definitely be done and compared to that for SVO-450.

Response to comment 3:

Thank you very much for the kind comments. Based on the reviewer's suggestion, we added TG/DTG data of Sr-V-O into Supplementary Information and form the new Supplementary Fig. 18 as below. We can know that the TG/DTG of Sr-V-O is similar to that of Ca-V-O, and the weight loss between 200 to 450 °C, which can be

corresponded to the evaporation of crystal water, is calculated to be 5.2%.

Supplementary Figure 18 | Characterization of the Sr-V-O nanowires. (a) XRD patterns of the Sr-V-O nanowire samples at different annealing condition. (b) TG/DTG curves of unsintered Sr-V-O nanowire sample at Ar atmosphere. (c) SEM image of the SVO-450. Inset is the EDS result of SVO-450.

We have added some discussions about the TG/DTG of Sr-V-O in page 14 in the manuscript. The discussion is as follows:

The TG curve shows a weight loss of 5.2% between 200 to 450 °C, corresponding to the evaporation of crystal water (Supplementary Fig. 18b).

Respond to Reviewer #4:

The paper entitled "Alkaline earth metal vanadates as new sodium-ion battery anodes" reports the electrochemical performance and reaction mechanism of CaV_4O_9 nanowires for sodium ion batteries. It is remarkable that CaV_4O_9 nanowires exhibited stable capacity retention over 1600 cycles. However, this paper is not appropriate for the publication in *Nature Communications* because of lack of novelty.

Our response:

We thank the reviewer for the time and efforts throughout the process together with the critical comments about our manuscript. But we respectfully disagree with the reviewer as regards the lack of novelty of this work, and moreover, we strongly believe that our finding, characterizations, analyses and the proposed mechanism described in the manuscript lay a strong foundation for the further research and development of alkaline earth metal vanadates as electrodes in energy storage, and thus meet the criteria for publication in *Nature Communications*.

We welcome the opportunity to address and clarify the issues raised in the reviewer's report, and believe that the additional experiments and revisions carried out to address the reviewer's comments substantially strengthen our revised manuscript. Our responses to the points raised in the report are as follows:

1a. CaV_4O_9 is irreversibly decomposed into CaO and VO_2 through a conversion reaction in the first cycle, and Na^+ ions are inserted into VO_2 in the subsequent cycles. This means that VO_2 is a real active material, although CaV_4O_9 is a starting material. However, VO_2 is well known as anode materials for sodium ion batteries, and many groups studied their electrochemical performance and reaction mechanism.[1,2] In other words, CaV_4O_9 is not a practically new and novel material and can be considered as a mixture of CaO and VO_2 , although CaV_4O_9 is first examined for sodium ion batteries in this paper.

[1] Steffen Hart et al., Vanadium-based polyoxometalate as new material for sodium-ion battery anodes, *Journal of Power Sources*, 288, (2015) 270–277

[2] C. Didier et al., Electrochemical Na-Deintercalation from NaVO_2 , *Electrochem. Solid-State Lett.* 14, (2011) A75-A78

Response to comment 1a:

We respectfully disagree with the reviewer's comments.

First, reviewer-4 lists two publications and claims that " VO_2 is well known as anode materials for sodium ion batteries, and many groups studied their electrochemical performance and reaction mechanism". However, Reference [1] is about the $\text{Na}_6\text{V}_{10}\text{O}_{28}\cdot 16\text{H}_2\text{O}$, and reference [2] is about the investigation of the NaVO_2 crystal as cathode in the voltage range from 1.5-3.0 V, both of which are not about VO_2 and are totally different with our work. In addition, as far as we know, the good electrochemical properties, superior Na-storage performance and detailed reaction mechanism of CaV_4O_9 displayed in this work have never been reported in previous reports. For VO_2 , although several publications reported its electrochemical performance as SIB anode, the detailed reaction mechanism still has not been fully

studied.

Second, reviewer-4 thinks that CaV_4O_9 is not a novel material due to its decomposition to VO_2 and CaO , and VO_2 is the real active material. However, the reviewer has ignored the effects of the *in situ* generated CaO nanograins and the greatly improved electrochemical performance of CaV_4O_9 compared to VO_2 .

As displayed in Figure 3c and 3d, CaV_4O_9 exhibits a much superior rate capability and cycling stability than VO_2 . Based on our analyses, the *in situ* generated CaO nanograins play a crucial role on the substantially improved electrochemical performance. As displayed in Figure 2b and 2c, the distribution of Ca ions in the CaV_4O_9 crystal structure is very uniform at atomic scale level, which indicates that the generated CaO are also distributed uniformly in the whole matrix after Na^+ insertion. The evenly distributed CaO nanograins can effectively inhibit the agglomeration and grain-growth of the NaVO_2 and VO_2 nanograins during charge/discharge process. And the small grain size of NaVO_2 and VO_2 (~ 5 nm, as demonstrated by TEM results in Figure 5) benefits the reversible desodiation and sodiation processes in the subsequent cycles, as indicated by the *ab initio* calculations. To cite the text from original manuscript in page 11:

Next, considering the small size of generated NaVO_2 and VO_2 nanograins (~ 5 nm, observed in TEM characterization), we used three models of (001), (010) and (110) surfaces (Supplementary Figure 16a-c) to study the surface effect of NaVO_2 nanograins on the Na dissociation. The calculated dissociation energies of Na from the (001), (010) and (110) surfaces were -0.23 eV, 0.50 eV and 0.41 eV, respectively. These small values indicate that the Na dissociation from all the three surfaces is feasible and highly reversible, suggesting a reversible conversion between NaVO_2 and VO_2 nanograins as equation (2) in charge-discharge processes. Moreover, a model of NaVO_2 crystal (Supplementary Figure 16d) was also applied to calculate the dissociation energy of Na . In this case, the average energy reached a large value of 5.63 eV, indicating the conversion from NaVO_2 crystal to VO_2 is hard to be realized. Thus, the reaction of equation (2) is related to the particle size, and the generated nanograins during charge/discharge are beneficial to the conversion between NaVO_2 and VO_2 . Besides, these results highlight the importance of the generated CaO nanograins, which inhibit the agglomeration and grain-growth of the NaVO_2 and VO_2 nanograins during charge/discharge process, and keep the electrodes with high reversibility.

Therefore, the *in situ* generated CaO nanograins effectively improve the cycling reversibility of the electrodes. Both the small size (<5 nm for CaO , and ~ 5 nm for NaVO_2 and VO_2) and the uniform distribution (at atomic scale level) of CaO and $\text{NaVO}_2/\text{VO}_2$ account for the superior electrochemical performance; these features and the produced effects derived from CaV_4O_9 can be hardly achieved by simple mixing of CaO and VO_2 . Therefore, CaV_4O_9 cannot be considered as simple mixture of VO_2 and CaO , but a new electrode material with intrinsic good electrochemical properties.

Third, to further reflect the unique and intrinsic properties derived from CaV_4O_9 , we added *ex situ* TEM data of VO_2 at sodiation state after 300 cycles for comparison. The results are shown below:

Supplementary Figure 15 | Ex situ TEM characterization of VO₂-450 at the sodiation state after 300 cycles. (a) TEM image, (b) SAED pattern, (c) HRTEM image.

TEM image shows that the nanowire morphology keeps integrated for VO₂ after 300 cycles. SAED pattern indicates the generation of NaVO₂ at sodiation state. But different from Figure 5f and 5g, strong diffraction spots rather than diffraction rings were observed, indicating the large grain size of NaVO₂. From the HRTEM image, the distinct nanograins were observed with grain size of ~20 nm, which is much larger than that derived from CaV₄O₉. Based on the *ab initio* calculation results in the manuscript, the large grain size of NaVO₂ is not beneficial to the Na dissociation, which will lead to poor reversibility of the sodiation/desodiation process. These new results together with our previous analyses further explain why VO₂ has a poor cycling stability even though its small volume change during charge/discharge, and further highlight the important role of the *in situ* generated CaO derived from CaV₄O₉.

We have added these new results as Supplementary Figure 15 and added corresponding discussions in page 11 as follows:

The *ex situ* TEM measurement was also performed on VO₂ nanowires at the sodiation state after 300 cycles. The nanowire morphology also keeps integrated according to the TEM image (Supplementary Fig. 15a). The SAED pattern and the HRTEM image manifest the formation of NaVO₂ (JCPDS: 00-044-0342) at the sodiation state (Supplementary Fig. 15b,c). However, different from that observed at the sodiation state of CaV₄O₉, the SAED pattern displays strong diffraction spots rather than diffraction rings (Supplementary Fig. 15b), indicating that the generated NaVO₂ display a large grain size. From the HRTEM image, the distinct nanograins were observed with grain size of ~20 nm (Supplementary Fig. 15c), much larger than those derived from CaV₄O₉.

Based on the new results and our previous analyses, we revised Figure 7 as below to further clarify the unique property of CaV₄O₉ and the effect of generated CaO nanograins (about which the reviewer also raised comments below), and the corresponding discussions in page 14 were revised as follows:

Based on our comprehensive analyses, the reaction mechanism of the CaV₄O₉ was proposed, as shown in Fig. 7a. The Na⁺ insertion into CaV₄O₉ crystal results in the irreversible structural dissociation and amorphization, meanwhile, NaVO₂ and CaO nanograins are generated and dispersed uniformly in the amorphous matrix. In the

desodiation process, the NaVO_2 nanograins are converted to VO_2 nanograins, and CaO act as a spectator. The superior electrochemical performance of CaV_4O_9 can be attributed to the following aspects: First, both the pristine CaV_4O_9 and the generated VO_2 have good electric conductivity, which ensures good conductance of the nanowires throughout the whole discharge/charge process. Second, the structural dissociation and amorphization process leads to increase of the active surface area, which is beneficial to the ion diffusion kinetics. The good conductance and the increased active surface area contribute to the good rate capability. Besides, the *in situ* generated and uniformly distributed CaO nanograins produce a self-preserving effect in the whole subsequent cycles, which effectively inhibit the agglomeration and grain-growth of the active components, resulting in the high reversibility of the conversion between NaVO_2 and VO_2 nanograins in the desodiation/sodiation process. Finally, the entire sodiation/desodiation process produces a small volume change of less than 10%, which further ensures the good cycling stability. In the case of VO_2 crystal (Fig. 7b), due to the absence of the self-preserving effect from CaO , the generated NaVO_2 nanograins after the Na^+ insertion tend to agglomerate into relatively large sized particles as the cycling continues, which results in the poor reversibility of the subsequent desodiation/sodiation processes, and then the observed capacity fading.

Figure 7 | Illustration of the reaction mechanism of (a) CaV_4O_9 and (b) VO_2 during initial sodiation and subsequent cycles.

1b. In addition, CuV_2O_6 showing a similar conversion reaction mechanism has already introduced as an anode for lithium ion batteries.[3,4]

[3] P. Prahastini et al., A novel attempt for employing brannerite type copper vanadate as an anode for lithium rechargeable batteries, J Mater Sci: Mater Electron 27, (2016) 3292–3297

[4] Kangning Zhao, Graphene Oxide Wrapped Amorphous Copper Vanadium Oxide with Enhanced Capacitive Behavior for High-Rate and Long-Life Lithium-Ion Battery Anodes, *Adv. Sci.* 2, (2015) 1500154

Response to comment 1b:

We believe the provided references are substantially different from our present work.

First, reference [3] provided by the reviewer just reports the synthesis and Li-storage performance of CuV_2O_6 , no results or data were displayed about the reaction mechanism. The detailed reaction products of CuV_2O_6 are not clear according to reference [3]. But for our work, the reaction mechanism of CaV_4O_9 and the effect of CaO were carefully studied through *in situ/ex situ* XRD, XPS and TEM measurements together with *ab initio* calculations. Different from the derived CaO from CaV_4O_9 , the possible CuO/Cu₂O/Cu derived from CuV_2O_6 is electrochemically active, whether they can bring a self-preserving effect like CaO is also not clear. Therefore, the provided reference [3] is totally different from our work.

Second, reference [4] is focused on the investigation of the enhanced capacitive behavior of amorphous CuVO. And metallic Cu nanoparticles were demonstrated to be generated, which was proposed to increase the conductivity and benefit to the capacitive behavior. But for this work, CaV_4O_9 was demonstrated to generate CaO, which inhibits the agglomeration and grain-growth of the active nanograins, and thus improves the reversibility of the desodiation/sodiation process. Both the reaction mechanism and the effects are different.

Third, the provided reference [3] and [4] focus on the lithium storage performance, but our work focus on the sodium storage performance and the reaction mechanism. The research direction is also different; lithium storage is not identical to sodium storage.

To sum up, we strongly believe that our work is substantially different from the provided references and previous reports. Our finding, characterization and the demonstrated good electrochemical performance and clarified reaction mechanism, indicate the great potential of CaV_4O_9 and other alkaline earth metal vanadates for SIB anodes. We strongly believe that this work will bring positive impacts on the development of SIBs, and may open new directions in the exploration of electrode materials for energy storages.

2. The specific capacity and average redox potential of CaV_4O_9 are about 300 mAh/g and about 1 V vs. Na/Na^+ . Considering the energy density of sodium ion batteries, CaV_4O_9 is not an attractive anode material because of its poor specific capacity and high average redox potential.

Response to comment 2:

We thank the reviewer for the critical comment, but we respectfully disagree with the reviewer's viewpoints. As we known, large scale energy storage applications such as smart grids, are the main development targets for sodium ion batteries, in which case, safety, power density and cycle life are more important than energy density (Refs. 2, 3, revised manuscript). The average potential of ~1 V may reduce the energy density of the full cells compared with hard carbon, but it will greatly improve the safety of the

batteries. As pointed out by Ceder's group (Ref. 24, revised manuscript), the average potential of ~ 1 V is a moderate one with a comprehensive consideration of safety and energy density.

The capacity of CaV_4O_9 is demonstrated to be ~ 300 mAh g^{-1} , which is actually higher than most of the Ti-based anodes, as displayed in Figure 1 in the manuscript. And together with the good rate capability and excellent cycling stability, we believe that CaV_4O_9 is a promising anode material for sodium-ion batteries.

Besides, an example that we would like to take is $\text{Li}_4\text{Ti}_5\text{O}_{12}$ as anode material in Li-ion batteries, whose redox voltage and practical capacity are ~ 1.5 V and ~ 175 mAh g^{-1} , respectively, but now it has been commercialized as anode for Li-ion batteries. Thus, considering the good rate capability, excellent cycling stability, applicable capacity and moderate voltage, we believe that CaV_4O_9 will have a bright prospect for sodium ion battery anode.

3. The authors insisted that the volume change of CaV_4O_9 is small ($\sim 10\%$) as compared to P, Sn, and Sb. However, this comparison is not fair, because the specific capacities of P, Sn, and Sb are much larger than that of CaV_4O_9 . The volume change of CaV_4O_9 should be compared with that of VO_2 or hard carbon delivering similar reversible capacity (about 300 mAh/g). For hard carbon, there is no volume change during charge and discharge.

Response to comment 3:

We thank the reviewer for the comment. Volume change is an important property for the electrodes during charge/discharge, especially for anodes. For a material researcher, one of the most important tasks is to reveal the fundamental properties of new materials. Considering that this is the first report about CaV_4O_9 as SIB anodes, our intention is to reflect the intrinsic property (small volume change) of this material during sodiation/desodiation, and explain the good cycling stability we observed. To cite the text in page 13 from original submission:

Thus, the volume change in the entire discharge/charge process is very small (less than 10% based on a conservative estimation), which explains why the nanowire morphology is retained in both the discharge and charge state, even after 300 cycles. The small volume change during the sodiation/desodiation process essentially benefits the electrochemical stability²⁷.

We think both the expressions and explanations about the volume change are reasonable and appropriate, and will lay a foundation for the understanding and further investigation about this material or other alkaline earth metal vanadates.

For VO_2 or hard carbon, we agree that both their volume changes are also small (less than 10%). It can be expected that the volume changes of CaV_4O_9 , VO_2 and hard carbon will not show much difference, and the difference will not be a decisive factor for the electrochemical performance.

4. The authors explained that the improved cycle performance of CaV_4O_9 is attributed to the buffering role of CaO. However, the volume change of VO_2 itself is not large during charge and discharge. Therefore, it is difficult to find what the buffering role of CaO is. Of course, it is okay to say that the buffering role of CaO contributes to

capacity retention of metal oxides such as CaO+SnO showing large volume change, as the authors cited. Therefore, the buffering role of CaO should be intensively studied, because this phenomenon is one of the most important aspects of CaV₄O₉.

Response to comment 4:

We thank the reviewer very much for the comments. As our response to comment 1, the effect of CaO not only represents the buffering of the volume change, more importantly, it represents the inhibition of the agglomeration and grain-growth of the active nanograins. Based on the reviewer's comments, we have changed the "self-buffering effect" into "self-preserving effect" in the manuscript to better display the important role of the generated CaO.

Also as our response to comment 1, we added *ex situ* TEM data of VO₂ at sodiation state after 300 cycles (as below) for comparison. SAED pattern indicates the generation of NaVO₂ at sodiation state. But different from Figure 5f and 5g, strong diffraction spots rather than diffraction rings were observed, indicating the large grain size of NaVO₂. From the HRTEM image, the distinct nanograins were observed with grain size of ~20 nm, which is much larger than that derived from CaV₄O₉.

Supplementary Figure 15 | *Ex situ* TEM characterization of VO₂-450 at the sodiation state after 300 cycles. (a) TEM image, (b) SAED pattern, (c) HRTEM image.

Based on the new results and our previous analyses, we revised Figure 7 as below to further clarify the effect of generated CaO nanograins, and the corresponding discussions in page 14 were revised as follows:

Besides, the *in situ* generated and uniformly distributed CaO nanograins produces a self-preserving effect in the whole subsequent cycles, which effectively inhibit the agglomeration and grain-growth of the active components, resulting in the high reversibility of the conversion between NaVO₂ and VO₂ nanograins in desodiation/sodiation process. Finally, the entire sodiation/desodiation process produces a small volume change of less than 10%, which further ensures the good cycling stability. In the case of VO₂ crystal (Fig. 7b), due to the absence of the self-preserving effect from CaO, the generated NaVO₂ nanograins after Na⁺ insertion tend to agglomerate into relatively large sized particles as the cycling continue, which results in the poor reversibility of the subsequent desodiation/sodiation processes, and then the observed capacity fading.

Figure 7 | Illustration of the reaction mechanism of (a) CaV₄O₉ and (b) VO₂ during initial sodiation and subsequent cycles.

5a. CaV₄O₉ delivered higher specific capacity than its theoretical value, and the authors explained that the extra capacity is attributed to the Na⁺ storage in the amorphous region of CaV₄O₉ or the residual water or OH groups on the surface of the nanowires. However, these claims were not supported by any evidence, although the extra capacity is an important abnormal behavior of CaV₄O₉.

Response to comment 5a:

We thank the reviewer for the comments. The phenomenon of extra capacity beyond the theoretical capacity is actually observed in many conversion based materials. Our explanation about the extra capacity is mainly based on the previous reports, as we cited in the original manuscript. Chae et al. (Ref. 42, revised manuscript) has reported the much higher capacity of amorphous V₂O₅ than crystalline V₂O₅, and based on computational simulations and experimental results, the vacant sites, which are highly populated in amorphous phase, are believed as the additional active sites to provide extra capacity. Besides, many other works also reported similar results^{R1,R2}. Thus, it is reasonable that we attribute one of the factors of additional capacity to the amorphous region.

For the other factor, the residual water, which is proposed to the reason why CVO-450 has a higher capacity but an inferior cycling stability compared to CVO-550. Grey's group (Ref. 56, revised manuscript) has clearly revealed that the residual water can provide additional capacity in the low voltage, but the residual water also leads to the consumption and decomposition of the electrolyte, which is harmful to the long-term stability of the batteries. For this work, the nanowire sample contain large amount of crystal water (~6.2%) before annealing, as shown in Supplementary Fig.4-6. Based on the XRD results and FT-IR spectra in Supplementary Fig. 5 (as shown below), the

crystal water did not fully evaporate after being annealed at 400 °C (CVO-400). Thus, it is reasonable that minor residual water will also exist in the sample annealed at 450 °C (CVO-450).

Supplementary Figure 5 | (a) XRD patterns of the Ca-V-O nanowire samples at different annealing condition. (b–d) FT-IR spectra of Ca-V-O nanowire samples before annealing (b), sintered at 400 °C for 5 h (c) and sintered at 450 °C for 8 h (d), respectively. (For the FT-IR spectra, the paraffin oil was used as the dispersant rather than KBr to avoid the moisture absorption during the test.)

To further give evidence to this point, we tested the performance of CVO-400, and make comparison with CVO-450 and CVO-550. The results have been added as Supplementary Figure 17 as below. It is known that, as the annealing temperature increases from 400, to 450 and 550 °C, the capacity decreases, but the cycling stability increases greatly. This trend can be well explained by the effect of water. For CVO-400, considerable water still exists in the sample, as displayed by the XRD and FT-IR results in Supplementary Fig.5, which leads to the higher initial capacity but poor stability due to the decomposition of the electrolyte. For CVO-450, only minor residual water exists, thus it exhibits an intermediate performance both for capacity and cycling stability. For CVO-550, the crystal water was completely removed, and it displays a much better stability with a lower capacity.

Supplementary Figure 17 | Comparison of specific capacity and cycling stability of CVO-400, CVO-450 and CVO-550 at current density of 1000 mA g^{-1} .

To further clarify this point, we revised the corresponding discussions in page 12 as follows:

The capacity of CVO-450 is higher than that of CVO-550 at different current densities. The extra capacity can be attributed to the residual water in the nanowires. As described before, the nanowire samples contain considerable crystal water ($\sim 6.2\%$) before annealing (Supplementary Fig. 4-6). As indicated by the XRD and FT-IR results (Supplementary Fig. 5a and 5c), the crystal water was still not fully removed for CVO-400 (annealed at $400 \text{ }^\circ\text{C}$); thus, it is reasonable that there is minor residual water on the CVO-450. As reported by Grey's group⁵⁶, residual water or OH groups can act as a major source of additional capacity. However, residual water also leads to the consumption and decomposition of the electrolyte and does not benefit the long-term cycle life of the batteries, which explains why CVO-450 has inferior cycling stability compared to CVO-550 (Fig. 3d). To further confirm this point, CVO-400 was also tested as electrode and compared with CVO-450 and CVO-550. As expected, the CVO-400 exhibits higher initial capacity but much poorer cycling stability (Supplementary Fig. 17).

Reference:

- R1. Ku, J. H. *et al.* Reversible lithium storage with high mobility at structural defects in amorphous molybdenum dioxide electrode. *Adv. Funct. Mater.* 22, 3658-3664 (2012).
 R2. Niu, C. J. *et al.* Carbon-supported and nanosheets-assembled vanadium oxide microspheres for stable lithium ion battery anodes, *Nano Research*, 31, 52-57 (2016).

5b. Also, the authors used large amount of carbon additive (about 30 wt.% of active material) for the electrode preparation, and carbon additive is known to deliver the reversible capacity of about 250 mAh/g. This means that about 70 mAh/g of specific capacity can be contributed from carbon additive. This amount of specific capacity is similar to that of the extra capacity of CaV_4O_9 in this report.

Response to comment 5b:

We thank the reviewer for the comments. The working electrodes were prepared by mixing 70% active material, 20% acetylene black and 10% carboxyl methyl cellulose (CMC) binder, which is a commonly used ratio. And as far as we known, the acetylene black is different from hard carbon, which cannot deliver reversible capacity as high as $\sim 250 \text{ mAh g}^{-1}$. To confirm this point, we tested the capacity of pure acetylene black that we used (the electrodes were prepared by mixing 90% acetylene black and 10% CMC binder), and the results are shown below. It is known that the reversible capacity is $\sim 90 \text{ mAh g}^{-1}$, which means that contributed capacity from acetylene black is only $\sim 20 \text{ mAh g}^{-1}$. Therefore, carbon additive is not the main contributor for the extra capacity as we observed.

6. The authors insisted that the good electrical conductivity of CaV_4O_9 contributes to improve electrochemical performance. However, CaV_4O_9 is irreversibly decomposed into CaO and VO_2 through conversion reaction in the first cycle. Therefore, little CaV_4O_9 exists in the subsequent cycles, indicating that CaV_4O_9 can not contribute the improved electrochemical performance.

Response to comment 6:

We thank the reviewer for the critical comment. The good conductivity is an intrinsic property of CaV_4O_9 . As our Response to comment 3, from the point of fundamental research, we think it is necessary to indicate this property due to the first investigation of CaV_4O_9 as electrode materials. And it is well known that good electric conductivity is very important for the electrode materials. It is true that CaV_4O_9 decomposed into CaO and VO_2 , however, the good conductivity still benefits to the initial conversion reaction, and the efficient initial conversion benefits to the whole subsequent cycles. We can expect that if the original material has a poor conductivity, the initial conversion may hardly come up, and so does the subsequent cycles.

Finally, we still thank the reviewer-4 very much for the time and efforts during the review process, even though some critical or even negative comments on our work were given. These critical comments have provided us an opportunity to make further revisions and substantial improvements on this manuscript. We believe that the

additional experiments and results together with the revisions have greatly improved the quality of this work, which further strengthens our confidence about this paper. Besides, reviewer-2 and reviewer-3 have given very positive comments and high evaluations on our work. Therefore, we strongly believe that this work is significantly different from the previous reports, and the results presented in this manuscript will attract great interests of the broad readerships and hence make positive impacts on the development of SIBs.

We now believe that the comments from reviewer-4 have been carefully considered and addressed, and the ambiguities have been clearly clarified. We hope reviewer-4 could give more objective and positive assessments on this work.

Response Letter

Respond to Reviewer #1:

The revised manuscript is improvement over previous version. However, wild claims need to be removed before it can be accepted for publication:

Our response:

We thank the reviewer very much for the positive assessment together with the additional suggestions. We have made further revisions in the manuscript according to the reviewer's recommendation. We believe these revisions substantially addressed the reviewer's concerns, and the revised manuscript now reaches the level for publication. Our responses are as follows:

1. Abstract: Remove the statement: "which result in its long-term cycling stability over 1,600 cycles." The test for 1600 cycles was performed at high current which means short time. There is no evidence that the material is stable over long term. **Our response:**

We thank the reviewer for the reminding. After carefully considering the reviewer's comments, we agree that a more appropriate statement should be provided, and we have revised the abstract cautiously. The corresponding revision is as follows:

Here, we identify alkaline earth metal vanadates as new and promising anodes for SIBs. The prepared CaV_4O_9 nanowires possess intrinsically high electronic conductivity ($>100 \text{ S cm}^{-1}$), small volume change ($<10\%$) and a self-preserving effect, which result in a superior cycling and rate performance and an applicable reversible

capacity (>300 mAh g⁻¹), with an average voltage of approximately 1.0 V.

2. There is no reason to believe that the material is scalable and hence claims regarding large scale energy storage need to be removed: Abstract: "large-scale energy storage".

Our response:

We thank the reviewer for the comment. Based on the reviewer's recommendation, we have removed "large-scale energy storage" in abstract and revised the corresponding sentence as follows:

The abundance of sodium resources indicates the great potential of sodium-ion batteries (SIBs) as emerging energy storage devices.

3. Figure 3b. Conductivity of the powder is more important than individual nanowire. Four point conductivity of the nanowire powder (or pressed pellet) is recommended.

Our response:

We thank the reviewer for the suggestion. Following the reviewer's recommendation, we pressed the CVO-450 nanowire powders into a small wafer (as shown below), and tested its conductivity based on four point method. The conductivity of the pressed pellet was tested to be $10^{-2} \sim 10^{-3} \text{ S cm}^{-1}$. The value is lower than that tested by single nanowire device due to the existence of contact resistance or interface resistance in the pressed pellet.

Generally speaking, regarding to conductivity, the value tested by single nanowire device reveals the intrinsic conductivity of the material, which is more important for the scientific research (like this work). And the value tested by four point method is a reflection of large amounts of powder, which may be more important for the technical adjustment in practical production.

Respond to Reviewer #2:

Reviewer #2 (Remarks to the Author):

The authors have issued all the concerns, so I think it is suitable to publish the paper in NC.

Our response:

We thank the reviewer every much for the positive and kind comments and the strong support on our work.

Respond to Reviewer #3:

Reviewer #3 (Remarks to the Author):

I am satisfied with the response provided by the authors to my comments and recommend this manuscript for publication in "Nature Communications".

Our response:

We are glad that the reviewer is satisfied with our response, and we thank you very much for your time and efforts throughout the process, and also thank you for the positive assessment on our work.

Respond to Reviewer #4:

Reviewer #4 (Remarks to the Author):

The authors have addressed all of my previous concerns, and their revisions have substantially improved the manuscript. I agree that the proposed conversion reaction mechanism of CaV_4O_9 is new and interesting. However, some issues are not still clear.

Our response:

We are glad that the reviewer is satisfied with our previous revisions, and thank you very much for the positive assessments on our work. We welcome the additional comments from the reviewer to discuss the important points about this work. We have made further revisions in the manuscript, and believe the concerns from the reviewer have been carefully addressed and clarified. Our responses to the points are as follows:

1. CaV_4O_9 showed low reversible capacity (less than 350 mAh/g), poor coulombic efficiency for the initial cycle (55~70%), and high redox potential (0~3 V). These properties result in poor energy density of full cells. Based on that $\text{Li}_4\text{Ti}_5\text{O}_{12}$ with a high redox potential (~1.5 V vs. Li/Li^+) is commercialized, the authors insisted that the high redox potential of CaV_4O_9 is not a problem for commercialization. However, CaV_4O_9 shows slopping voltage profiles, while $\text{Li}_4\text{Ti}_5\text{O}_{12}$ has a plateau profile at ~1.5 V vs. Li/Li^+ . In the other words, the reversible capacity of CaV_4O_9 delivered in the voltage range of 1.5~3.0 V is useless, because the working voltage of full cells is too low to operate a device when the redox potential of anode is between 1.5~3.0 V. This indicates that the practical reversible capacity of CaV_4O_9 as anode is only about 200 mAh/g. In addition, the poor coulombic efficiency of the initial cycle (55~70%) will further decrease the energy density of full cells. Therefore, CaV_4O_9 is not practically promising, although the cycle performance of CaV_4O_9 is excellent.

Our response:

We thank the reviewer for the critical but valuable comments together with the analyses and discussion about the energy density.

First, for this manuscript, we believe the main merits are our fundamental analyses and discussion about the intrinsic properties and reaction mechanism of CaV_4O_9 as SIB anode. The obtained scientific insights will lay a foundation for the further research about alkaline earth metal vanadates in energy storage, which will bring positive impacts for the development of SIBs or other energy storage devices. As to the practical application of CaV_4O_9 , we agree that it of course has a long way to go. And whether it can be used in certain fields finally, just let time tell.

Second, we believe not all case has a high demand in energy density for batteries. For sodium ion batteries, it is well known that its developing target is mainly for stationary energy storage (the serving time is generally 5~10 years) due to the abundant resource of Na, in which case the long cycle life is believed to be more important than energy density.

Third, even though we take it that the practical reversible capacity of CaV_4O_9 is ~ 200 mAh g^{-1} (not include the capacity between 1.5~3 V) as the reviewer points out, it is still higher than $\text{Li}_4\text{Ti}_5\text{O}_{12}$ (~ 170 mAh g^{-1}). Besides, the average voltage of CaV_4O_9 is lower than that of $\text{Li}_4\text{Ti}_5\text{O}_{12}$ (~ 1.5 V). In these considerations, the energy density of the batteries based on CaV_4O_9 as the anode is comparable to or even higher than that based on $\text{Li}_4\text{Ti}_5\text{O}_{12}$. Considering that $\text{Li}_4\text{Ti}_5\text{O}_{12}$ has realized the practical application, we believe the application of CaV_4O_9 will not be limited by the energy density.

2. The role of CaO is not still clear. Alloy materials with high volume changes ($>100\%$) during charging and discharging are known to show poor cycle performance because of pulverization and repetitive electrolyte decomposition during cycling. However, NaVO_2 shows small volume changes ($\sim 10\%$) during charging and discharging, suggesting that the pulverization will not occur during cycling. This means that NaVO_2 does not require the buffering or preserving role of CaO.

Our response:

We thank the reviewer for the important comments. It is true that NaVO₂ shows small volume changes, but the degradation mechanism of NaVO₂ or VO₂ as anode is different from that of alloy materials. Based on our analyses, the role of CaO, in a word, is to preserve the small size (~5 nm) of generated NaVO₂ nanograins and keep the conversion between NaVO₂ and VO₂ in a high reversibility (as displayed in Figure 7 in the manuscript).

Based on the *ab initio* calculations in the manuscript, the reversibility of conversion between NaVO₂ and VO₂ is highly relevant to the particle size. As discussed in page 11 in the manuscript, the dissociation energy of Na from NaVO₂ crystal model (represent large sized NaVO₂ particle) reaches a large value of 5.63 eV, but which is no more than 0.50 eV for NaVO₂ surfaces model (represent small NaVO₂ nanograins). These results indicate that small sized NaVO₂ nanograins are beneficial to the reversible conversion between NaVO₂ and VO₂, and then benefit to the cycling performance.

For CaV₄O₉, the Na⁺ insertion leads to the generation of NaVO₂ and CaO, and the inactive CaO will effectively prevent the NaVO₂ nanograins from agglomerating and keep the nanograins in a small size (~5 nm, as demonstrated in Figure 5) during sodiation/desodiation, and then preserve the high reversibility of the conversion reaction. But for pristine VO₂, due to the absence of the self-preserving effect from CaO, the generated NaVO₂ nanograins tend to agglomerate into relatively large sized particles (~20 nm, as demonstrated in Supplementary Figure 15) as the cycling continues, which results in the poor reversibility of the subsequent desodiation/sodiation processes, and then the observed capacity fading.

To further clarify the role of CaO, we revised the corresponding discussions in the revised manuscript.

In page 12, the discussion has been revised as follows:

Thus, the reaction of equation (2) is influenced by the particle size, and the small generated nanograins (~5 nm) derived from CaV₄O₉ are beneficial to the conversion between NaVO₂ and VO₂. Besides, these results highlight the importance of the

generated CaO, which prevent the NaVO₂ nanograins from agglomerating and preserve the small size of the active components, and then keep the electrodes with high reversibility.

In page 14, the discussion has been revised as follows:

Besides, the *in situ* generated and uniformly distributed CaO nanograins produce a self-preserving effect in the whole subsequent cycles, which effectively inhibit the agglomeration of the active components, and then preserve the high reversibility of the conversion between NaVO₂ and VO₂ nanograins in the desodiation/sodiation process.

Response Letter

Respond to Reviewer #1:

Reviewer #1 (Remarks to the Author):

Comments have been addressed to satisfaction.

Our Response:

We thank the reviewer very much for approving the publication of our work, and also thank you for your time and efforts throughout the review process.

Respond to Reviewer #4:

Reviewer #4 (Remarks to the Author):

The authors have addressed essentially all of my previous concerns, and it is appropriate for the publication in Nature Communications.

Our Response:

We thank the reviewer very much for the positive assessment on our work, and also thank you for your time and efforts throughout the process.